# Complete chloroplast genomes of *Asparagus aethiopicus* L., A. *densiflorus* (Kunth) Jessop 'Myers', and A. *cochinchinensis* (Lour.) Merr.: Comparative and phylogenetic analysis with congenerics

**Kwan-Ho Wong**[1,2], **Bobby Lim-Ho Kong**[2,3], **Tin-Yan Siu**[1], **Hoi-Yan Wu**[3], **Grace Wing-Chiu But**[2], **Pang-Chui Shaw**[2,3,4], **David Tai-Wai Lau**[1,3]*

**1** Shiu-Ying Hu Herbarium, School of Life Sciences, The Chinese University of Hong Kong, Hong Kong Special Administrative Region, the People's Republic of China, **2** School of Life Sciences, The Chinese University of Hong Kong, Hong Kong Special Administrative Region, the People's Republic of China, **3** Li Dak Sum Yip Yio Chin R & D Centre for Chinese Medicine, The Chinese University of Hong Kong, Hong Kong Special Administrative Region, the People's Republic of China, **4** State Key Laboratory of Research on Bioactivities and Clinical Applications of Medicinal Plants (The Chinese University of Hong Kong) and Institute of Chinese Medicine, The Chinese University of Hong Kong, Hong Kong Special Administrative Region, the People's Republic of China

* lautaiwai@cuhk.edu.hk

**Data Availability Statement:** The data that support the findings of this study are openly available in

## Abstract

*Asparagus* species are widely used for medicinal, horticultural, and culinary purposes. Complete chloroplast DNA (cpDNA) genomes of three *Asparagus* specimens collected in Hong Kong—*A. aethiopicus*, A. *densiflorus* 'Myers', and A. *cochinchinensis*—were *de novo* assembled using Illumina sequencing. Their sizes ranged from 157,069 to 157,319 bp, with a total guanine–cytosine content of 37.5%. Structurally, a large single copy (84,598–85,350 bp) and a small single copy (18,677–18,685 bp) were separated by a pair of inverted repeats (26,518–26,573 bp). In total, 136 genes were annotated for *A. aethiopicus* and A. *densiflorus* 'Myers'; these included 90 mRNA, 38 tRNA, and 8 rRNA genes. Further, 132 genes, including 87 mRNA, 37 tRNA, and 8 rRNA genes, were annotated for *A. cochinchinensis*. For comparative and phylogenetic analysis, we included NCBI data for four congenerics, *A. setaceus*, A. *racemosus*, A. *schoberioides*, and A. *officinalis*. The gene content, order, and genome structure were relatively conserved among the genomes studied. There were similarities in simple sequence repeats in terms of repeat type, sequence complementarity, and cpDNA partition distribution. A. *densiflorus* 'Myers' had distinctive long sequence repeats in terms of their quantity, type, and length-interval frequency. Divergence hotspots, with nucleotide diversity (Pi) $\geq$ 0.015, were identified in five genomic regions: *accD-psaI*, *ccsA*, *trnS-trnG*, *ycf1*, and *ndhC-trnV*. Here, we summarise the historical changes in the generic subdivision of *Asparagus*. Our phylogenetic analysis, which also elucidates the nomenclatural complexity of *A. aethiopicus* and A. *densiflorus* 'Myers', further supports their close phylogenetic relationship. The findings are consistent with prior generic subdivisions, except for the placement of *A. racemosus*, which requires further study. These *de novo* assembled

GenBank (https://www.ncbi.nlm.nih.gov) with the accession number MZ337394, MZ337395 & MZ424304.

**Funding:** The research work was supported by a donation fund from Wu Jieh Yee Charitable Foundation Limited. The fund has no formal grant number. The funders had no role in study design, data collection and analysis, decision to publish, or preparation of the manuscript.

**Competing interests:** The authors have declared that no competing interests exist.

cpDNA genomes contribute valuable genomic resources and help to elucidate *Asparagus* taxonomy.

## Introduction

*Asparagus*, a genus with ca. 300 species [1–6], originated in southern Africa, particularly in the Cape of Good Hope. Some members are now distributed throughout tropical Africa, Eurasia, and Australia [4–11], mostly in arid and sub-arid regions [4–6]. *Asparagus* species have evolved their characteristic morphology as an adaptation to drought and arid environments [2, 4]. Their "true leaves" have been reduced to scales or spines, with the stem-derived organs ("cladodes") performing photosynthesis [2, 7, 8, 12, 13]. Cladode shape is variable, ranging from acicular, filiform, linear to cordate [2–4, 6, 11, 14–17]. Most species store nutrients and water in rhizomes or root tubers [2, 15–17].

   *Asparagus* species are commercially important worldwide [2, 7, 9, 10, 15, 18, 19], and many are widely used, particularly in medicinal, culinary, and horticultural applications. Here, we first summarise the anthropocentric uses and environmental impacts of some *Asparagus* species and then elucidate the complexity on generic subdivisions and nomenclature of the studied *Asparagus* species.

### Medicinal application

Many *Asparagus* species have medicinal value [19–29]. The root tubers of *A. cochinchinensis* (Lour.) Merr., 'Tiandong' in Traditional Chinese Medicine, are renowned for their therapeutical functions in nourishing yin, moistening dryness, clearing the heat and engendering fluid [30, 31]. *A. officinalis* L. [20–22, 24–27], *A. setaceus* (Kunth) Jessop [20], *A. filicinus* Buch.-Ham. ex D. Don [24, 28], *A. racemosus* Willd. [19, 21, 25, 27, 29], and *A. schoberioides* Kunth [29] have been used as herbal drugs in different regions for various functions. Root tubers of *A. filicinus* are used as adulterants of Stemonae Radix to cure tracheitis, pneumonia, coughing, and whooping cough [32–36]. In South African, several *Asparagus* species have been used to treat pulmonary tuberculosis, gonorrhoea, and infertility, while some *Asparagus* species have been used as charm to increase fertility, ensure victory, or fight against witchcraft [21].

### Culinary application

*Asparagus* species are an important culinary resource. Although young shoots of *A. officinalis* L., garden asparagus, are widely sold as a vegetable [1–4, 10, 27, 37], its gene pool is relatively limited [38–40]. It is susceptible to multiple biotic and abiotic stresses, including *Fusarium* rot [41, 42], *Puccinia asparagi* rust [43, 44], purple spot caused by *Stemphylium* [45–47], and stem blight caused by *Phomopsis asparagi* [48], negatively affecting its production and economic value. Attempts to cross *A. officinalis* with its wild relatives, to enhance tolerance to drought, disease, salinity, and acidity [49], have revealed that dioecious, but not monoecious, species could hybridize with it [44, 46, 50–54].

   Young shoots of *A. acutifolius* L., *A. aphyllus* L., and *A. albus* L. are also eaten as vegetables [55]. The fruits of *A. racemosus* are edible [56].

### Horticultural application

Owing to their distinct morphology, *Asparagus* species, including *A. setaceus*, *A. aethiopicus* L., and *A. densiflorus* (Kunth) Jessop 'Myers', have been widely used as ornamental plants

[1, 3, 57]. *The European Garden Flora* [57], published in 1986, mentions 24 *Asparagus* species, including *A. setaceus*, *A. aethiopicus*, *A. officinalis*, *A. densiflorus*, *A. filicinus*, *A. asparagoides* (L.) Druce, *A. falcatus* L., and *A. racemosus*. *The New Royal Horticultural Society Dictionary of Gardening* [3], published in 1992, reports the same number of species.

The xeromorphic adaptations of *Asparagus* species are beneficial to the establishment of "Xeroscaping" [58–60], a kind of landscaping that minimises the need for irrigation. The *Pictorial Guide to Plant Resources for Skyrise Greenery in Hong Kong* (Developmental Bureau of the Hong Kong Special Administrative Region Government) [61–63] recommends three *Asparagus* species—*A. cochinchinensis*, *A. aethiopicus* (recorded as *A. densiflorus* 'Sprengeri'), and *A. densiflorus* 'Myers'—as skyrise greenery.

## Environmental impacts

Global cultivation of *Asparagus* species has promoted the invasiveness of the species, particularly of the horticultural species. The berries of *Asparagus* species are a food source for birds, further promoting their seed dispersal [64]. The invasiveness of *Asparagus* species has been widely recorded in, for instance, Australia [10, 65–67] and the USA [60, 64].

## Genus-level taxonomical complexity

Linnaeus first described the genus *Asparagus* in 1753 [68]. Since the publication of the genus *Mysiphyllum* by Willdenow in 1808 [14], generic circumscription of the genus *Asparagus* have been disputed [67, 69, 70]. Based on morphological characters, taxonomists have divided the genus *Asparagus sensu lato* into three genera: genus *Protasparagus* [16, 17, 72] (also known as *Asparagopsis*, an illegitimate homonym [71, 73]); genus *Asparagus sensu stricto* [16, 17, 71–73]; and genus *Myrsiphyllum* [16, 17, 71–73]. The genus *Asparagus sensu lato* has also been divided into three subgenera (subgenus *Asparagopsis*, *Euasparagus*, and *Myrsiphyllum*) [7], or even multiple sections or races [7–9, 15] (S1 Fig). The key morphological characteristics for generic subdivision include the sexual strategy (monoecy or dioecy), perianth segments (free or connate), filaments (free or connate into column), number of ovules per locule (2 or more), cladode shape and arrangement, and presence or absence of spines.

Later evidences and analysis revealed that these subdivisions were not clear-cut. While Malcomber and Demissew [69] advocated to combine these subdivisions into two subgenera under the genus *Asparagus* (subgenus *Asparagus* and subgenus *Myrsiphyllum*), Fellingham and Meyer [70] suggested eliminating the generic subdivisions. It has been stated that "until the phylogenetic relationships within *Asparagus* are investigated in more details, the recognition of any infrageneric groups is problematic" [4].

Norup *et al.* [6] utilised chloroplast and nuclear genome barcode regions (*trnH-psbA*, *trnD-trnT*, 3′ *ndhF*, and *PHYC*) in their classification: using 211 accessions representing 119 species, they divided the genus *Asparagus* into six major clades and multiple subclades (S1 Fig).

## Species and infraspecific taxonomical complexity

Only one *Asparagus* species, *A. cochinchinensis*, has been recorded as native to Hong Kong. Exotic species that are common in Hong Kong include Sprenger's asparagus (*A. aethiopicus*), foxtail asparagus (*A. densiflorus* 'Myers'), lace fern (*A. setaceus*), and garden asparagus (*A. officinalis*). The nomenclature of Sprenger's asparagus and foxtail asparagus is controversial.

**Sprenger's asparagus.** The nomenclature of this species is unclear. In 1890, Regel published the name *Asparagus sprengeri* based on cultivated plants growing in Natal, Africa [74, 75]. The epithet *sprengeri* is after Mr. Sprenger, the co-owner of Dammann & Co., which produced this cultivated plant. The name *A. sprengeri* Regel was adopted by Baker (1875) [7] and

Geiner (1919) [9]. In 1966, Jessop [15] synonymised *A. sprengeri* Regel under the new combination *A. densiflorus* (Kunth) Jessop, based on morphology and geographical distribution. Since then, it has been commonly recorded as *A. densiflorus*, based on Jessop [1, 3, 5, 10, 11, 57, 64, 70]. It has even been considered a cultivar ('Sprengeri') [1, 57, 64] or a group (the "Sprengeri group") [3] of *A. densiflorus*.

The name *A. aethiopicus* dates from 1767 (S1 Table), when Linnaeus published it in *Species Plantarum* [68]. Eighty-three years later, Kunth [71] transferred the species to the genus *Asparagopsis*. It was later subdivided under the genus *Asparagus* by Baker (in 1875 and 1896) [7, 8] and Jessop (in 1966) [15]. In 1983, Obermeyer [16] transferred it to a new genus *Protasparagus*, because *Asparagopsis* is an illegitimate homonym. Malcomber and Demissew [69] combined the genera *Protasparagus* and *Asparagus* into genus *Asparagus* subgenus *Asparagus* in 1992. Fellingham and Meyer [70], however, cancelled all generic subdivisions three years later, moving it back to the genus *Asparagus*.

*Aspararagopsis aethiopica* (and later *Asparagus aethiopicus*) and *Asparagopsis densiflora* were adopted in parallel for 116 years, from 1850 to 1965. In 1996, Jessop [15] classified both species in the genus *Asparagus* (S1 Table). However, these species are considered highly variable [4, 15]. According to Green (1989) [76], Jessop (1966) [15], Judd (2001) [4], and Straley and Utech (2004) [77], the growth habit of *A. aethiopicus* is more variable, ranging from arching herbs of ca. 1 m in length to scrambling climbers of ca. 7 m in length. In 1986, Green [76] disagreed with Jessop's treatment [15] of *A. sprengeri* as *A. densiflorus*, which is a small-sized species. Green ascribed Jessop's treatment to the omission of *A. densiflorus* from Regel's protologue in *Gartenflora* [75] and to misidentification of cultivated materials, which rarely reach their full potential size as potted plants. Following Judd in 2001 [4], Straley and Utech, in *Flora of North America North of Mexico* (2004) [77], also adopted *A. aethiopicus* for Sprenger's asparagus, stating "*Asparagus densiflorus* (Kunth) Jessop has been misapplied to this species". They considered Sprenger's asparagus to be a cultivar, suggesting the combination as *A. aethiopicus* 'Sprengeri'. On the contrary, Conran, in *Horticultural Flora of South-eastern Australia* [78], treated it as "Sprengeri Group" of *A. aethiopicus*.

The voucher specimens of our research materials were authenticated based on the latest *Asparagus* monograph, *The Genus Asparagus in South Africa* [15], and the *Flora of Hong Kong* [79]. The voucher specimen of Sprenger's asparagus (K. H. Wong 109), collected in Hong Kong, fit the circumscription of *A. aethiopicus* L. in the monograph, based on their habitats, growth habit, and reproductive characteristics. Therefore, we have adopted *A. aethiopicus* L. for Sprenger's asparagus in this study.

**Foxtail asparagus.** This cultivated plant was named for its foxtail-like branches, which are in narrow cones, assembled by orderly branchlets, densely surrounding the main stem, and gradually elongating from the stem apex [1, 3, 57, 64, 80]. Because of its popularity as an ornamental plant of good performance, the cultivar was named *A. densiflorus* 'Myersii' in the Royal Horticultural Society's *Award of Garden Merit* list [81].

The first binomial name of foxtail asparagus, *Asparagus myersii*, was raised anonymously at an unknown time, while *Asparagopsis densiflora* was validly published in 1850 by Kunth (S1 Table) [71]. The species epithet was named after Mr. Meyers, a nurseryman from East London, for the introduction of this plant [82]. In 1966, Jessop [15] mentioned that *Asparagus myersii* Hort. "had never been validly published", treating it as *nomen nudum*. At that time, he combined Kunth's *Asparagopsis* into *Asparagus* L., deeming this cultivated plant to be a form of *A. densiflorus*. In 1976, this plant was recorded as *A. densiflorus* 'Myers' by L. H. Bailey Hortorium in *Hortus III*, [1], treating it as a cultivar of *A. densiflorus*. Since then, this taxonomic treatment has been widely accepted by many taxonomists, horticulturalists, and scientists [3–5, 11, 27, 57, 64, 83].

The spelling of this cultivar epithet occurs in several forms, including the Latin form 'Myersii' [57, 80, 81] derived from the species epithet of its *nomen nudum*, the non-Latin form 'Myers' [1, 4, 5, 11, 51, 64, 76, 83] and 'Meyers' [82, 84, 85]. According to Article 21.6 of the *International Code of Nomenclature of Cultivated Plants* (ICNCP), "*the epithet of any name in Latin form published before 1 January 1959, even if it is not validly published under the International Code of Nomenclature for Algae, Fungi and Plants (ICN), that meets the requirements for establishment as a cultivar name under this Code (Art. 27.1), may be used as the cultivar epithet, if the plants to which it was applied are now considered to represent a cultivar*" [86]. Because these spellings exhibited no ambiguous indication to the same *Asparagus* cultivar as foxtail asparagus, we follow the treatment of some taxonomists and scientists [1, 4, 5, 11, 51, 64, 76, 83], adopting *A. densiflorus* (Kunth) Jessop 'Myers' for foxtail asparagus throughout this study.

## Provocative molecular evidence: The complete chloroplast genome

Past technical limitations restricted the molecular evidence for classification to short genomic fragments. Technological advancements have made the acquisition of complete genomes, and especially chloroplast genomes, more practicable, affordable, and popular. The chloroplast genome, described as a super-barcode [87–89], is important in studying phylogeny and resolving taxonomical problems [89–92].

Prior to the availability of complete chloroplast DNA (cpDNA) genomes, construction of physical maps of *Asparagus* cpDNA was attempted via Southern hybridisation of total DNA [93, 94]. Lee *et al.* [93] estimated the length of the *A. officinalis* 'Mary Washington 500W' cpDNA genome at ca. 155 kb, with two inverted repeats (IRs) of 23 kb each, separated by a 90 kb large single copy (LSC) and a 19 kb small single copy (SSC). The same group constructed the physical maps of cpDNA for another seven *Asparagus* species, *A. schoberioides*, *A. cochinchinensis*, *A. plumosus*, *A. falcatus*, *A. aethiopicus* (recorded as *A sprengeri*), *A. virgatus*, and *A. asparagoides* [94]. Their results suggest close relationships between these eight species. Despite the high similarity among these species, the cpDNA of *A. falcatus*, *A. sprengeri*, and *A. asparagoides* showed gain of the HindIII restriction site and loss of the XhoI restriction sites. Nucleotide deletion in *rbcL* was detected in *A. cochinchinensis* cpDNA [94].

The first *Apsaragus* cpDNA genome (NC_034777.1 = KY364194.1) was reported by Sheng *et al.* in 2017 [95], who assembled and annotated the cpDNA genome of *A. officinalis* 'Atlas' (length 156,699 bp); this revealed a quadripartite structure, including a pair of IRs (26,531 bp each), separated by an 84,999 bp LSC and 18,638 bp SSC, very similar to those reported by Lee *et al.* [93]. In 2019, Li *et al.* [96] reported the cpDNA genome of *A. setaceus* (NC_047458.1 = MK950153.1) of 156,978 bp, also quadripartite, and with a pair of IRs (26,513 bp each) separated by 85,311 bp LSC and 18,641 bp SSC. The cpDNA genome of *A. setaceus* is similar to that of *A. officinalis* 'Atlas' in terms of structure, gene order, and GC content.

GenBank (National Center for Biotechnology Information; NCBI) currently contains the cpDNA genomes of eight *Asparagus* species: *A. officinalis* (NC_034777.1 = KY364194.1, MT712156.1, LN896355.1, LN896356.1, MT712153.1, MT712155.1, and MT712154.1), *A. setaceus* (NC_047458.1 = MK950153.1 and MT712152.1), *A. cochinchinensis* (MW970105.1 and MW447164.1), *A. densiflorus* (MT740250.1), *A. dauricus* (MT712151.1), *A. schoberioides* (NC_035969.1 = KX790361.1), *A. racemosus* (NC_047472.1 = MN736960.1), and *A. filicinus* (NC_046783.1 = MK920078.1). This constitutes a small fraction of the genus, leaving a large knowledge gap in the molecular study of *Asparagus*.

We therefore aimed to revisit the phylogenetic relationships between two nomenclaturally confusing species *A. aethiopicus* and *A. densiflorus* 'Myers', using complete cpDNA genomes. This information will be useful in crossbreeding programmes, environmental remediation,

and authentication of medicinal materials. Using Illumina sequencing, we *de novo*-assembled the complete chloroplast genomes of *A. aethiopicus*, *A. densiflorus* 'Myers', and *A. cochinchinensis*. We performed comparative and phylogenetic analysis, including congenerics, using four cpDNA genomes from GenBank: *A. officinalis* (NC_034777), *A. racemosus* (NC_047472), *A. schoberioides* (NC_035969), and *A. setaceus* (NC_047458). The intra-generic relationships among these seven species were examined and compared to previous generic subdivision. Our analysis helps to elucidate and resolve the taxonomic positions and nomenclature of *A. aethiopicus*, *A. densiflorus* 'Myers', and other congenerics.

## Materials and methods

### Ethics statement

This study was conducted in accordance with Hong Kong Special Administrative Region legislation. Sample collection did not negatively affect the environment in any way.

### Plant material and DNA extraction

Individuals of the studied species were collected from the Chinese University of Hong Kong (Table 1 and Fig 1). Fresh and healthy cladodes were stored at −80˚C in a freezer immediately after collection. Voucher specimens were deposited at the Shiu-Ying Hu Herbarium (herbarium code: CUHK).

Total genomic DNA was extracted from 0.2 g of frozen cladode using the DNeasy Plant Pro Kit (Qiagen Co., Hilden, Germany) according to the manufacturer's instructions. Prior to the sequencing conducted by Novogene Bioinformatic Technology Co. Ltd. (http://en.novogene. com/, Beijing, China), DNA quantity and quality were assessed using a NanoDrop Lite Spectrophotometer (Thermo Fisher Scientific, MA, USA) and 1% agarose gel electrophoresis, respectively.

### cpDNA genome sequencing, assembly, and annotation

A paired-end library with an insert-size of 150 bp was constructed and sequenced on a Nova-Seq 6000 platform (Illumina Inc. San Diego, CA, USA). Raw reads were quality-trimmed using CLC Assembly Cell 5.1.1 (CLC Inc., Denmark), with Phred < 33. The trimmed reads were assembled into contigs using the CLC *de novo* assembler. Gaps were filled using Gapcloser in SOAPdenovo 3.23 to form contigs, then retrieved and ordered using NUCmer 3.0 [97]. The ordered contigs were aligned against reference chloroplast genomes. Based on phylogenetic proximity, *A. setaceus* (NC_047458) was selected as the reference genome for *A. aethiopicus* and *A. densiflorus* 'Myers', whereas *A. schoberioides* (NC_035969) was used for *A. cochinchinensis*. The aligned contigs were assembled into a complete cpDNA genome for each species.

Gene annotation of cpDNA was performed on the GeSeq platform (https://chlorobox. mpimp-golm.mpg.de/geseq.html) [98] based on the GenBank chloroplast genomes. *A. aethiopicus* and *A. densiflorus* 'Myers' were annotated in reference to *A. setaceus* (NC_047458) and

**Table 1. Information about the *Asparagus* specimens deposited at the Shiu-Ying Hu Herbarium.**

| Species | Collector no. | Inventory no. | Sheet no. | GPS location |
|---|---|---|---|---|
| *Asparagus aethiopicus* L. | K. H. Wong 109 | CUSLSH2801 | CUHK05891 | 22.420786, 114.208312 |
| *Asparagus densiflorus* (Kunth) Jessop 'Myers' | K. H. Wong 092 | CUSLSH2773 | CUHK05890 | 22.419994, 114.207354 |
| *Asparagus cochinchinensis* (Lour.) Merr. | K. H. Wong 107 | CUSLSH2799 | CUHK05892 | 22.421524, 114.207135 |

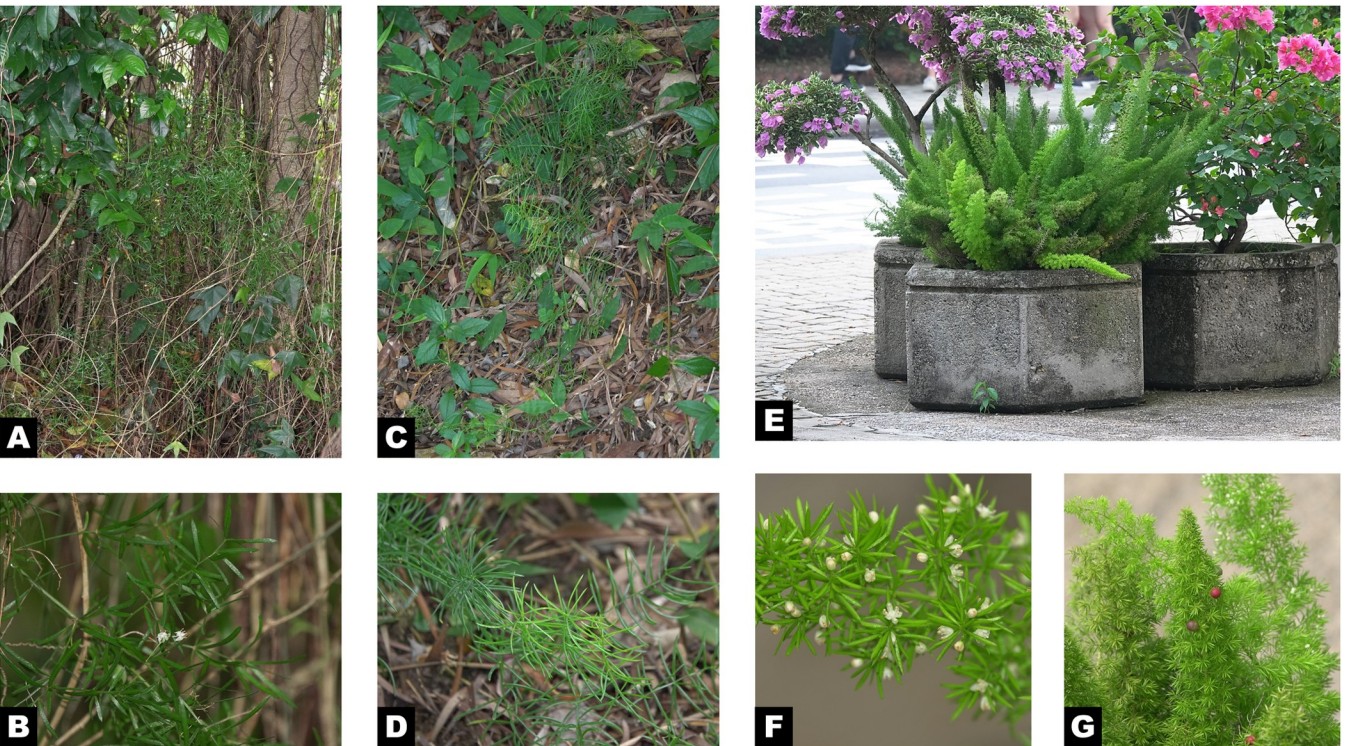

**Fig 1. Photos of three *Asparagus* plants collected at the Chinese University of Hong Kong.** A,B: *A. aethiopicus*. A. Plant climbing under *Ficus microcarpa* L. f. and twining with *Passiflora suberosa* L. B. Flowers and cladodes. C,D: *A. cochinchinensis*. C. Plant straggling on ground. D. Cladodes. E,F,G: *A. densiflorus* 'Myers'. E. Plant growing in a concrete pot. F. Flowers and cladodes. G. Fruits and branch apices.

*A. racemosus* (NC_047472), while *A. cochinchinensis* was annotated in reference to *A. schoberioides* Kunth (NC_035969) and *A. officinalis* L. (NC_034777). Manual adjustments, including editing the start and stop positions of genes and introns, were made where necessary. The circular genomic map was visualised by OrganellarGenomeDRAW (OGDRAW, https://chlorobox.mpimp-golm.mpg.de/OGDraw.html) [99]. The assembled and annotated chloroplast genomes of *A. aethiopicus*, *A. densiflorus* 'Myers', and *A. cochinchinensis* were submitted to GenBank (accession numbers MZ337394, MZ337395, and MZ424304, respectively).

## Repeat-sequence analysis

To compare the three newly assembled cpDNA genomes with chloroplast genomes of other *Asparagus* species, four cpDNA genomes (NC_034777, NC_047472, NC_035969, and NC_047458) were downloaded from GenBank. Repeat motifs, including simple sequence repeats (SSRs) and long sequence repeats (LSRs), were sequentially identified using the MIcroSAtellite identification tool (MISA, https://webblast.ipk-gatersleben.de/misa/index.php?action=1) [100] and REPuter (https://bibiserv.cebitec.uni-bielefeld.de/reputer) [101]. We screened for SSRs with at least 10, 5, 4, 3, 3, and 3 repeats, respectively, for mono-, di-, tri-, tetra-, penta-, and hexa-nucleotides. LSRs, including forward, reverse, complement, and palindromic sequences, were detected with a maximum computed repeat size of 50 bp and minimal repeat size of 30 bp.

## Comparative genome analysis

For structural comparison of the seven cpDNA genomes, we used mVISTA software (https://genome.lbl.gov/vista/mvista/submit.shtml) [102] to visualise the full alignment with

annotation, using the *A. aethiopicus* cpDNA genome as the reference. The shuffle-LAGAN alignment programme [103] was used.

To compare the size and type of IR border genes, IRScope (https://irscope.shinyapps.io/irapp/) [104] was used to visualise the junction sites of the seven cpDNA genomes. Junction gene positions and sizes were verified, and the diagram was redrawn manually.

To investigate divergence hotspots, the seven studied cpDNA genomes were first aligned using MAFFT 7 (https://mafft.cbrc.jp/alignment/server/) [105]. Sliding window analysis was conducted using DNA Sequence Polymorphism (DnaSP) 6.12.03 [106], which calculates the nucleotide diversity value (Pi) of the aligned cpDNA. The window length and step size were set to 600 and 200 bp, respectively.

### Phylogenetic analysis

The complete cpDNA genomes of the seven *Asparagus* species, with one outgroup species, *Hyacinthoides non-scripta* (L.) Chouard ex Rothm. (NC_046498), were used to construct maximum likelihood (ML) phylogenetic trees using the MEGA-X software [107], with 1000 bootstrap replicates for each tree. The best-fit model of nucleotide substitution, with the lowest Bayesian Information Criterion (BIC) scores, was calculated via ML model selection in MEGA-X. Respective trees were constructed from the aligned sequences of (i) complete cpDNA genome, (ii) protein coding (CDS) regions (excluding introns), (iii) LSC, (iv) SSC, and (v) IRs.

## Results

### *Asparagus* cpDNA genomes features

Illumina NovaSeq 6000 sequencing generated 3.2 Gb, 3.1 Gb, and 2.8 Gb raw data for *A. aethiopicus*, *A. densiflorus* 'Myers', and *A. cochinchinensis*, respectively. The cpDNA genomes were assembled with a coverage of 173x for *A. aethiopicus*, 164x for *A. densiflorus* 'Myers', and 381x for *A. cochinchinensis*.

The three newly assembled cpDNA genomes were relatively conserved in terms of length, gene order, gene content, and structure. The cpDNA genome of *A. densiflorus* 'Myers' was the largest (157,139 bp), followed by *A. aethiopicus* (157,069 bp), and *A. cochinchinensis* (156,319 bp; Table 2 and Fig 2). The cpDNA genomes exhibited the quadripartite structure typical of angiosperms. Their LSCs ranged from 84,598 to 85,350 bp in length and their IRs from 26,518 to 26,573 bp. The SSC was 18,677 bp for both *A. aethiopicus* and *A. densiflorus* 'Myers', and 18,685 bp for *A. cochinchinensis*.

Identical numbers and types of genes were annotated in *A. aethiopicus* and *A. densiflorus* 'Myers'. One hundred and thirty-six genes were successfully annotated, including 90 protein-coding (mRNA) genes, 38 transcription- and translation-related RNA (tRNA) genes, and 8 ribosomal RNA (rRNA) genes. For *A. cochinchinensis*, 132 genes were annotated, including 87 mRNA genes, 37 tRNA genes, and 8 rRNA genes. The genes were classified into three categories and 18 functions (Table 3).

The pseudogene *ycf1* occurred in *A. aethiopicus* and *A. densiflorus* 'Myers' but was not detected in *A. cochinchinensis*. *A. densiflorus* 'Myers' and *A. cochinchinensis* had 21 intron-containing genes, whereas *A. aethiopicus* had 20. All three cpDNA genomes had two genes comprising two introns (Table 4). For *A. aethiopicus* and *A. densiflorus* 'Myers', 20 genes were duplicated in IRs. In contrast, only 19 genes were duplicated in the IRs for *A. cochinchinensis*, because *ycf68* was absent from this genome.

The cpDNA genomes of the three species were comparable in terms of GC content (Table 2). In total, 37.5% of the GC bases were detected in all three cpDNA genomes; 35.4–

**Table 2. Summary on the cpDNA genome structure of the seven *Asparagus* species.**

|  | *A. aethiopicus* | *A. densiflorus* 'Myers' | *A. cochinchinensis* | *A. officinalis* | *A. racemosus* | *A. schoberioides* | *A. setaceus* |
|---|---|---|---|---|---|---|---|
| Accession no. | MZ337394 | MZ337395 | MZ424304 | NC_034777 | NC_047472 | NC_035969 | NC_047458 |
| **Total length (bp)** | **157,069** | **157,139** | **156,319** | **156,699** | **156,742** | **156,875** | **156,978** |
| LSC (bp) | 85,246 | 85,350 | 84,598 | 84,999 | 84,989 | 84,928 | 85,311 |
| SSC (bp) | 18,677 | 18,677 | 18,685 | 18,638 | 18,619 | 18,685 | 18,641 |
| IR (bp) | 26,573 | 26,556 | 26,518 | 26,531 | 26,567 | 26,631 | 26,513 |
| **Total number of genes** | **136** | **136** | **132** | **133** | **130** | **132** | **135** |
| mRNA | 90 | 90 | 87 | 88 | 86 | 88 | 90 |
| tRNA | 38 | 38 | 37 | 37 | 36 | 36 | 37 |
| rRNA | 8 | 8 | 8 | 8 | 8 | 8 | 8 |
| Pseudogene (Ψ) | 1[a] | 1[a] | 0 | 7[b] | 1[a] | 1[a] | 1[a] |
| 1-intron gene | 20 | 21 | 21 | 21 | 21 | 20 | 19 |
| 2-introns gene | 2 | 2 | 2 | 2 | 2 | 2 | 2 |
| **Total GC content (%)** | **37.49** | **37.49** | **37.54** | **37.59** | **37.55** | **37.57** | **37.48** |
| GC content in LSC (%) | 35.44 | 35.43 | 35.54 | 35.60 | 35.53 | 35.55 | 35.46 |
| GC content in SSC (%) | 31.30 | 31.31 | 31.38 | 31.50 | 31.43 | 31.51 | 31.45 |
| GC content in IR (%) | 42.94 | 42.93 | 42.90 | 42.92 | 42.92 | 42.93 | 42.85 |

[a] *ycf1*

[b] *ycf1*, *ycf15* (x2), *ycf68* (x2), *infA*, *rps19*.

35.5%, 31.3–31.4%, and 42.9% of the GC content was detected in LSCs, SSCs, and IRs, respectively. Among the three cpDNA genomes, *A. cochinchinensis* had the highest GC content (37.54%), with 35.54% in LSCs and 31.38% in SSCs, whereas *A. aethiopicus* had the highest IR GC content (42.94%).

## Simple sequence repeat analysis

The SSR number, type, content, and distribution were similar in the seven cpDNA genomes. The number of SSRs ranged from 80 (*A. schoberioides*) to 88 (*A. aethiopicus* and *A. officinalis*) (Fig 3).

Each cpDNA sample contained mono-, di-, tri-, or tetra-nucleotides. Three of the seven cpDNA genomes contained pentanucleotides, whereas the other four contained hexanucleotides. The most common class of SSRs was mononucleotides, ranging from 47 in *A. densiflorus* 'Myers' to 57 in *A. officinalis*. Dinucleotides were the second most common, ranging from 12 in *A. racemosus* to 15 in *A. aethiopicus* and *A. densiflorus* 'Myers'. Tetranucleotides were the third most common, ranging from 10 in *A. schoberioides* to 13 in *A. aethiopicus* and *A. densiflorus* 'Myers'. Trinucleotides repeats were the least common, with five each in *A. cochinchinensis*, *A. officinalis*, *A. racemosus*, and *A. schoberioides*, and seven each in the other species. One or two pentanucleotide or hexanucleotide repeats were found in each of the seven genomes.

Considering sequence complementarity, most of the SSRs were A/T (adenosine/thymine) repeats. ranging from 46 in *A. densiflorus* 'Myers' to 55 in *A. officinalis* (Fig 4). AT/AT repeats were the second most common, from 9 in *A. racemosus* to 12 in *A. aethiopicus* and *A. densiflorus* 'Myers'. AAAT/ATTT repeats were the third most common, at 4 in *A. officinalis*, 6 in *A. schoberioides*, and 7 in the other cpDNA genomes.

For the seven genomes, 87.59% of the SSRs comprised entirely adenosine and thymine, with at most 2 bp of guanine and cytosine in the GC-containing SSRs. The dominance of A/T

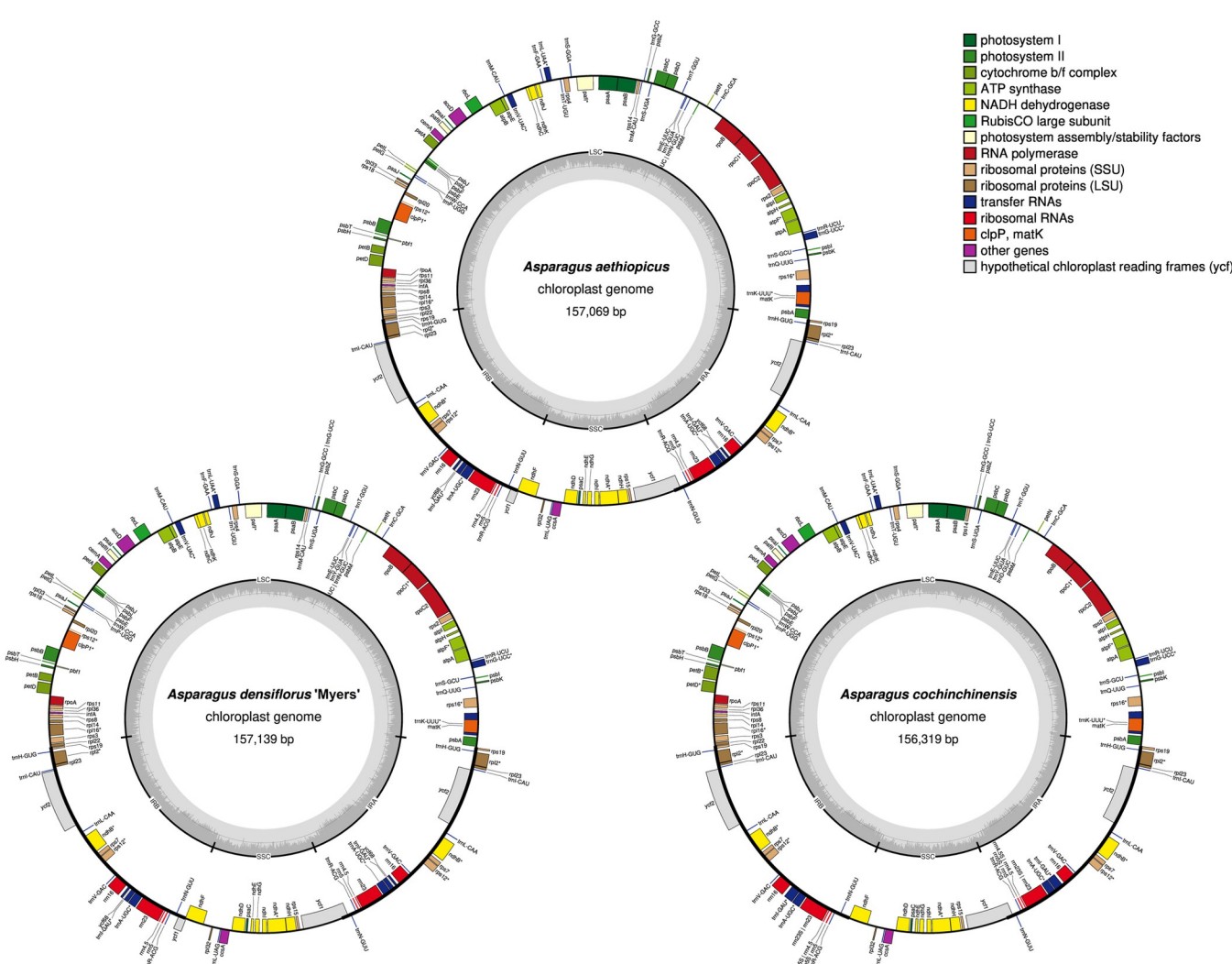

**Fig 2. Chloroplast genome map of *A. aethiopicus* L., *A. densiflorus* (Kunth) Jessop 'Myers', and *A. cochinchinensis* (Lour.) Merr.** Genes are colour-coded based on their functions shown in the key. Genes located outside of the outer circle are transcribed anticlockwise, while those inside are transcribed clockwise. In the inner circle, the gradient in dark grey represents GC content, whereas light grey represents AT content.

base pairs and low frequency of G/C base pairs in SSRs are consistent with the observations made by Sheng *et al.* [95].

The cpDNA genomes demonstrated similar proportional distributions of SSRs within the quadripartite structure (Fig 5), with most (ca. two-thirds) found in LSC regions and one-fifth and one-tenth, respectively, found in SSC and IR regions.

## Long sequence repeat analysis

The species differed significantly in the LSR analysis, particularly for *A. densiflorus* 'Myers' (Figs 6 and 7): for the other six genomes, there were 2 LSRs (*A. officinalis* and *A. schoberioides*) to 5 LSRs (*A. cochinchinensis*), whereas *A. densiflorus* 'Myers' had 34 LSRs, almost 10-fold the average in the others.

All four types of LSRs (forward, reverse, palindromic, and complement repeat) were detected. Notably, the genomes contained from 1 (*A. officinalis*) to 3 (*A. densiflorus* 'Myers')

**Table 3. Genes annotated in the complete cpDNA genomes of *A. aethiopicus* L., *A. densiflorus* (Kunth) Jessop 'Myers', and *A. cochinchinensis* (Lour.) Merr.**

| Gene category | Gene functions | Gene names |
|---|---|---|
| Photosynthesis-related genes | Rubisco | *rbcL* |
| | Photosystem I | *psaA, psaB, psaC, psaI, psaJ* |
| | Assembly/ stability of photosystem I | *pafI, pafII, pbf1* |
| | Photosystem II | *psbA, psbB, psbC, psbD, psbE, psbF, psbH, psbI, psbJ, psbK, psbL, psbM, psbT, psbZ* |
| | ATP synthase | *atpA, atpB, atpE, atpF, atpH, atpI* |
| | Cytochrome b/f complex | *petA, petB, petD, petG, petL, petN* |
| | Cytochrome c synthesis | *ccsA* |
| | NADPH dehydrogenase | *ndhA, ndhB[%], ndhC, ndhD, ndhE, ndhF, ndhG, ndhH, ndhI, ndhJ, ndhK* |
| Transcription- and translation-related genes | Transcription | *rpoA, rpoB, rpoC1, rpoC2* |
| | Ribosomal protein | *rpl2[%], rpl14, rpl16, rpl20, rpl22, rpl23[%], rpl32, rpl33, rpl36, rps2, rps3, rps4, rps7[%], rps8, rps11, rps12[%], rps14, rps15, rps16, rps18, rps19[%]* |
| | Translation initiation factor | *infA* |
| RNA genes | Ribosomal RNA | *rrn16[%], rrn23[%], rrn4.5[%], rrn5[%]* |
| | Transfer RNA | *trnA-UGC[%], trnC-GCA, trnE-UUC, trnF-GAA, trnG-GCC, trnG-UCC[\*], trnH-GUG[%], trnI-CAU[%], trnI-GAU[%], trnK-UUU, trnL-CAA[%], trnL-UAA, trnL-UAG, trnM-CAU[$], trnN-GUU[%], trnN-GUC, trnP-UGG, trnQ-UUG, trnR-ACG[%], trnR-UCU, trnS-GCU, trnS-GGA, trnS-UGA, trnT-GGU, trnT-UGU, trnV-GAC[%], trnV-UAC, trnW-CCA, trnY-GUA* |
| Miscellaneous group | Maturase | *matK* |
| | Inner membrane protein | *cemA* |
| | ATP-dependent protease | *clpP1* |
| | Acetyl-CoA carboxylase | *accD* |
| | Unknown functions | *ycf1[@], ycf2[%], ycf68[#]* |

[%] Duplicated in inverted repeat regions

[\*] Duplicated in large single copies of *A. densiflorus* 'Myers' and *A. cochinchinensis*; appeared once in *A. aethiopicus*

[$] Duplicated in large single copies of *A. aethiopicus* and *A. densiflorus* 'Myers'; appeared once in *A. cochinchinensis*

[@] *ycf1* was functional in all three species, but the *ycf1* pseudogene was absent from *A. cochinchinensis*

[#] Duplicated in inverted repeat regions of *A. aethiopicus* and *A. densiflorus* 'Myers'; absent from *A. cochinchinensis*.

types of LSRs. Palindromic repeats were the most common LSR type: of the 29 palindromic repeats, *A. densiflorus* 'Myers' had 17. Forward repeats were second, occurring in five of the species, excluding *A. officinalis* and *A. racemosus*. Of the 22 forward repeats, *A. densiflorus* 'Myers' had 16. *A. densiflorus* 'Myers' and *A. racemosus* had 1 reverse repeat and 1 complement repeat, respectively.

The minimum repeat size was set to 30 bp. The longest LSR detected by REPuter was 56 bp. LSRs were detected at lengths of 30, 31, 32, 33, 34, 35, 36, 38, 39, 46, 47, 49, 52, 54, and 56 bp. Fig 7 represents their frequencies in three intervals: (i) 30–39 bp, (ii) 40–49 bp, and (iii) 50–56 bp. LSRs of 30–39 bp and 50–56 bp occurred in all three species, whereas only *A. densiflorus* 'Myers' has LSRs of 40–49 bp (six, in total). LSRs of 30–39 bp were the most common, with 39 detected. *A. densiflorus* 'Myers' had the most in this class, at 26. Each of the three species had at least one 50–56 bp LSR, while *A. densiflorus* 'Myers' had two.

## Comparative genome analysis

The IR boundaries of the seven genomes were relatively conserved, with some minor variations (contractions and deletions) (Fig 8).

**Table 4. Intron-containing genes in the chloroplast genomes of seven *Asparagus* species.**

| | *A. aethiopicus* | *A. densiflorus* 'Myers' | *A. cochinchinensis* | *A. officinalis* | *A. racemosus* | *A. schoberioides* | *A. setaceus* | Location |
|---|---|---|---|---|---|---|---|---|
| Accession no. | MZ337394 | MZ337395 | MZ424304 | NC_034777 | NC_047472 | NC_035969 | NC_047458 | / |
| *trnK-UUU* | 0 | 1 | 1 | 1 | 1 | 1 | 0 | LSC |
| *rps16* | 1 | 1 | 1 | 1 | 1 | 1 | 1 | LSC |
| *trnG-UCC* [B] | 1 | 1 | 1 | 1 | 1 | 1 | ABS | LSC |
| *atpF* | 1 | 1 | 1 | 1 | 1 | 1 | 1 | LSC |
| *rpoC1* | 1 | 1 | 1 | 1 | 1 | 1 | 1 | LSC |
| *ycf3/ pafI* [C] | 2 | 2 | 2 | 2 | 2 | 2 | 2 | LSC |
| *trnL-UAA* | 1 | 1 | 1 | 1 | 1 | ABS | 1 | LSC |
| *trnV-UAC* | 1 | 1 | 1 | 1 | 1 | 1 | 1 | LSC |
| *clpP* | 2[A] | 2[A] | 2[A] | 2 | 2 | 2 | 2 | LSC |
| *petB* | 1 | 1 | 1 | 1 | 1 | 1 | 1 | LSC |
| *petD* | 1 | 1 | 1 | 1 | 1 | 1 | 1 | LSC |
| *rpl16* | 1 | 1 | 1 | 1 | 1 | 1 | 1 | LSC |
| *rpl2* *2 | 1 | 1 | 1 | 1 | 1 | 1 | 1 | $IR_A + IR_B$ |
| *ndhB* *2 | 1 | 1 | 1 | 1 | 1 | 1 | 1 | $IR_A + IR_B$ |
| *rps12* *2 | 1 | 1 | 1 | 1 | 1 | 1 | 1 | $IR_A + IR_B + LSC$ |
| *trnI-GAU* *2 | 1 | 1 | 1 | 1 | 1 | 1 | 1 | $IR_A + IR_B$ |
| *trnA-UGC* *2 | 1 | 1 | 1 | 1 | 1 | 1 | 1 | $IR_A + IR_B$ |
| *ndhA* | 1 | 1 | 1 | 1 | 1 | 1 | 1 | SSC |

0—No intron; 1–1 intron; 2–2 introns; ABS—Gene absent.

[A] Annotated as *clpP1*.

[B] Located in the region 9167–9994 bp; for NC 047458, *trnG-UCC*, at 36924–36994 bp, had no intron.

[C] *pafI* was annotated in *A. aethiopicus*, *A. densiflorus* 'Myers', and *A. cochinchinensis*.

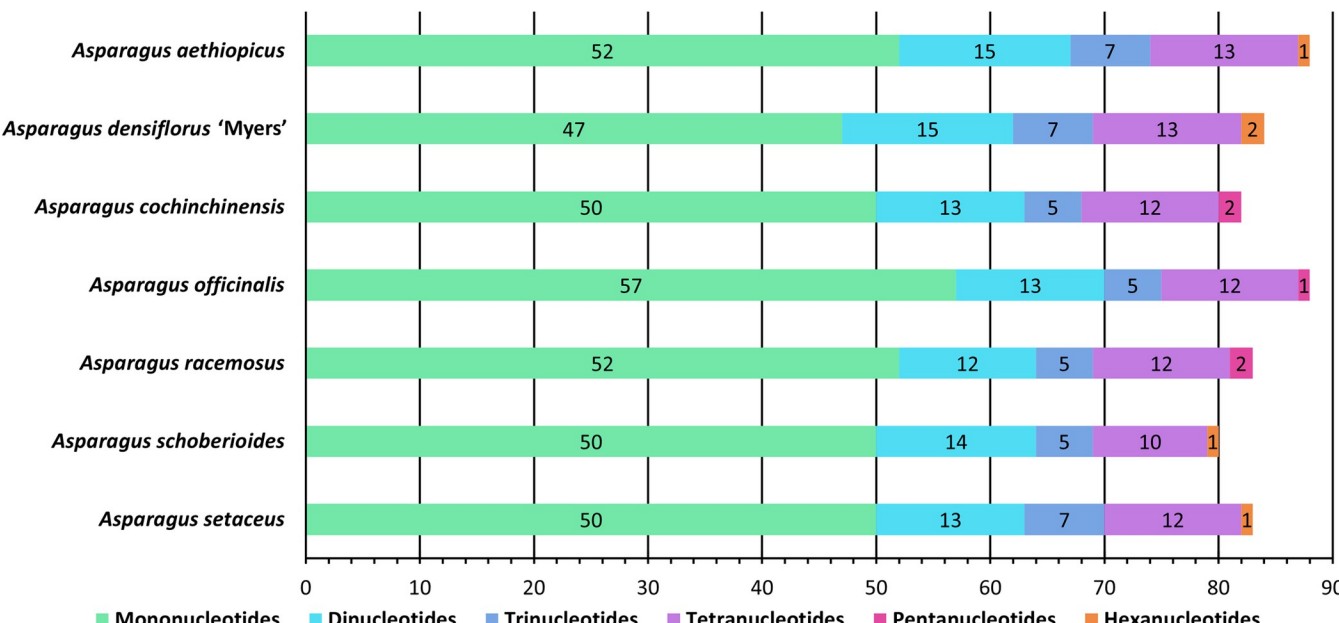

**Fig 3. Simple sequence repeat class distribution.**

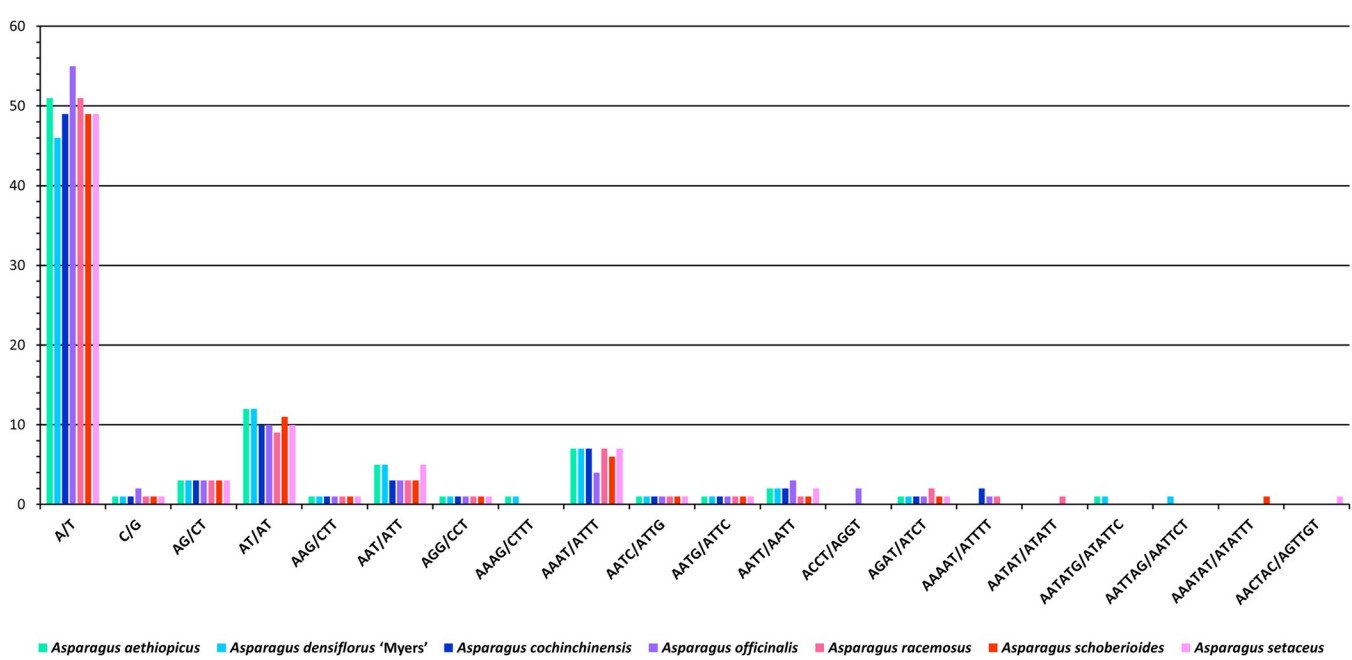

**Fig 4. Simple sequence repeat frequency related to sequence complementarity.**

In the LSC/IR$_B$ border, *rpl22* extended into the LSC by 2–5 bp from the junction, for all species except *A. cochinchinensis*, in which it extended it by 24 bp. For *A. officinalis*, *rpl22* was 360 bp long, 3 bp shorter than in the others. *rps19* in the IR$_B$ also exhibited variation, with lengths

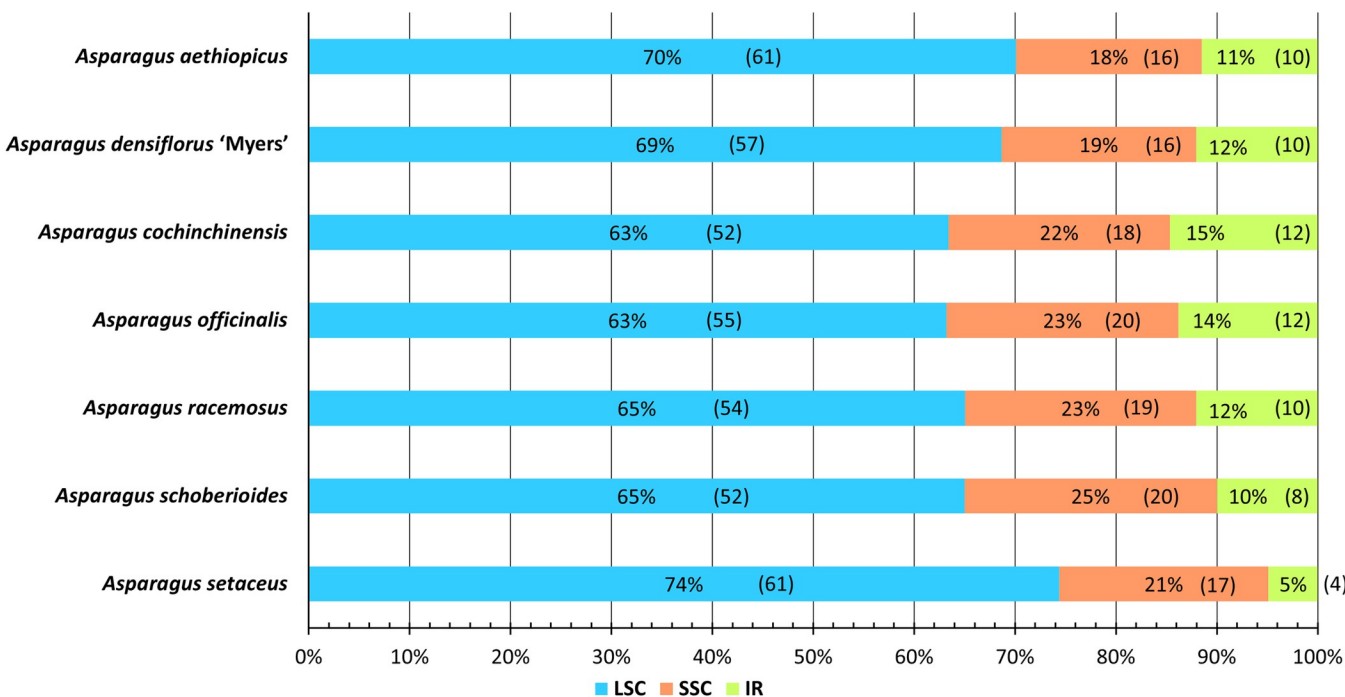

**Fig 5. Simple sequence repeat distribution in the quadripartite cpDNA structure.** The percentages for each region are shown in the middle of each bar. The numbers in brackets are the actual numbers of SSRs distributed in the indicated cpDNA regions.

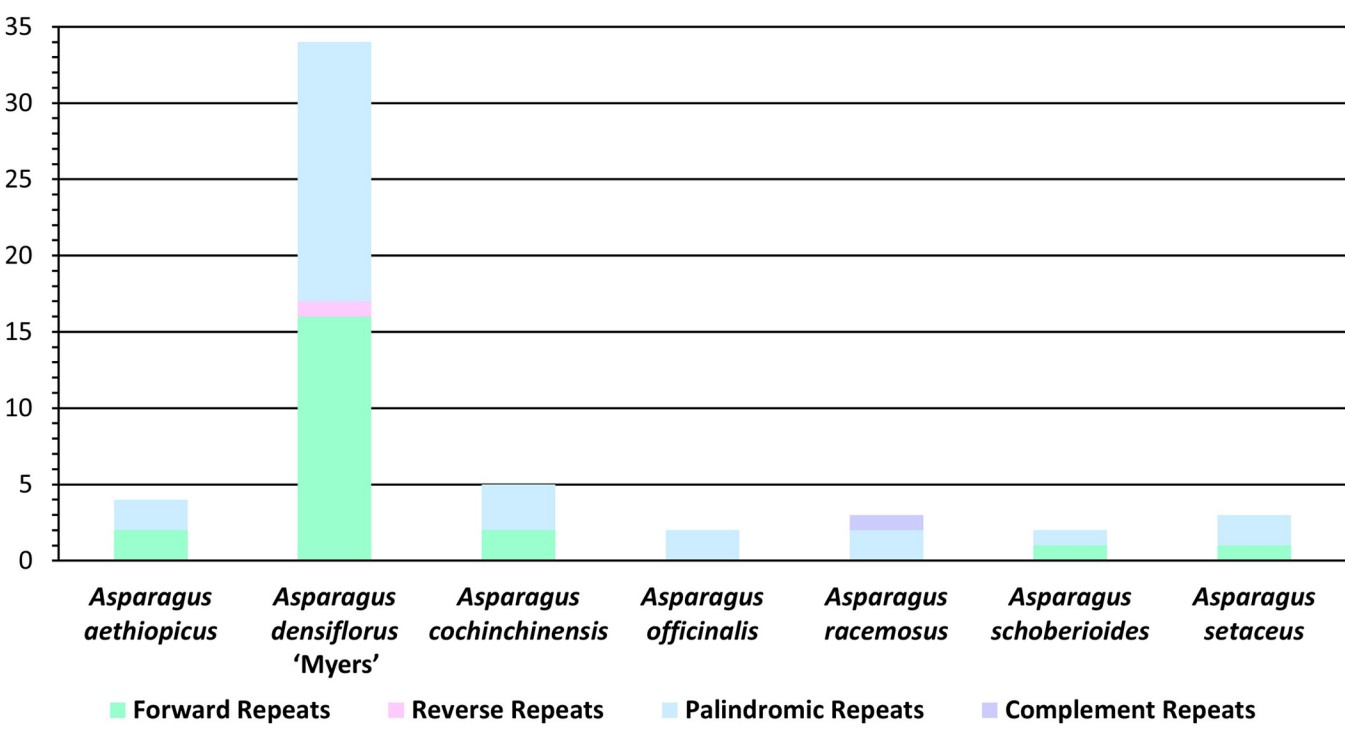

**Fig 6. Types of long sequence repeats.**

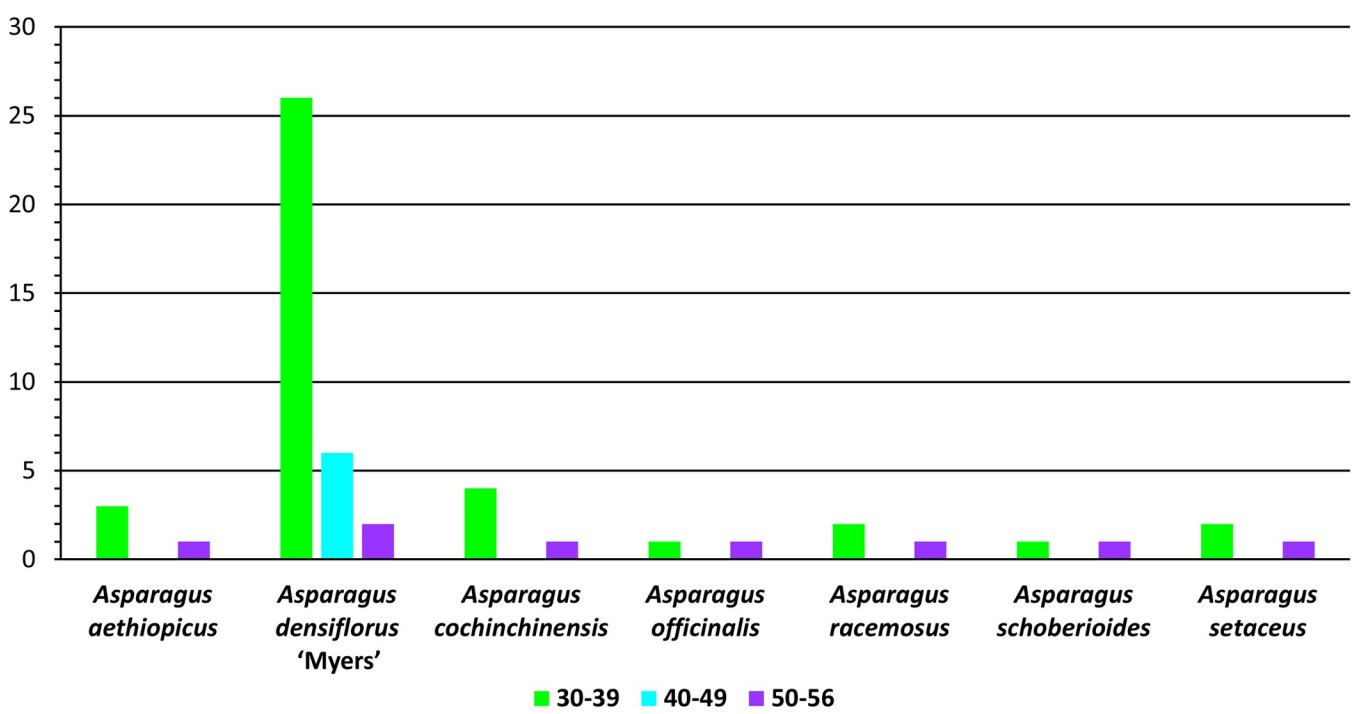

**Fig 7. Frequency of long sequence repeats in specified length intervals.**

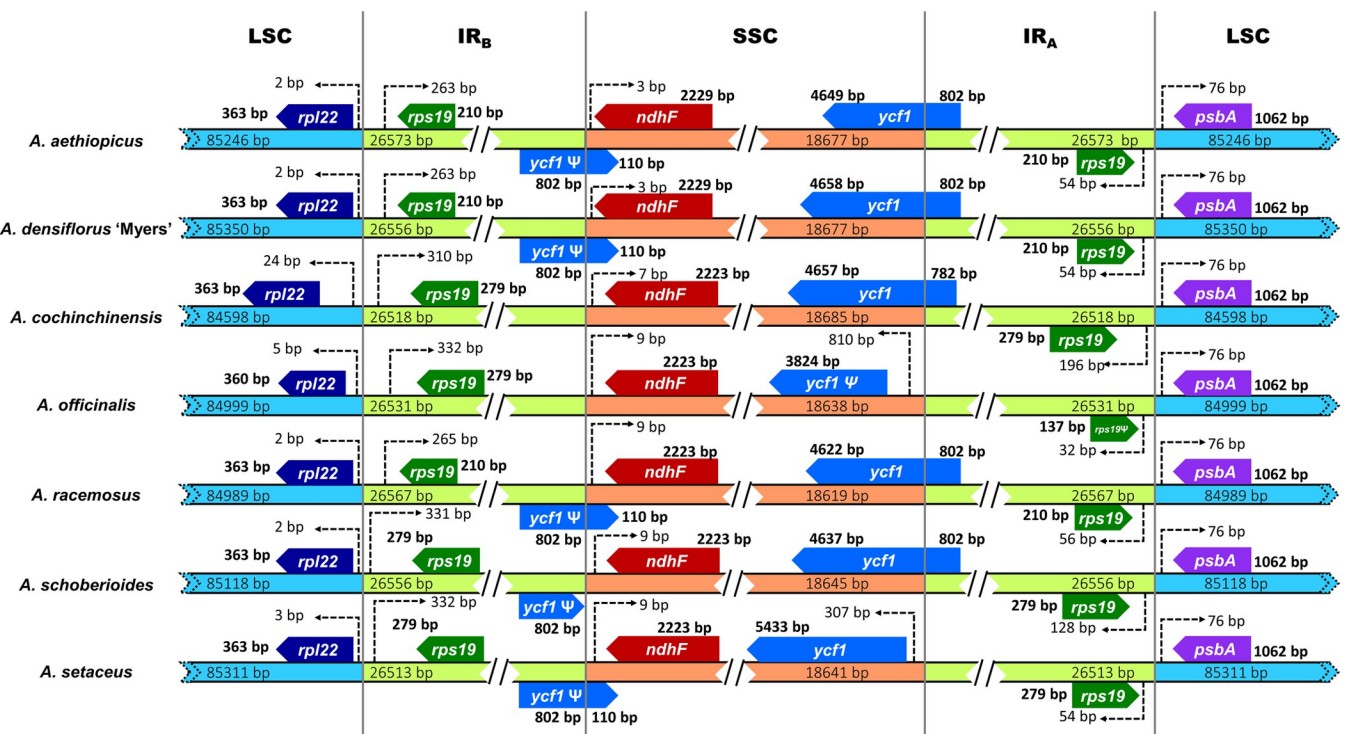

**Fig 8. Large single copy (LSC), small single copy (SSC), and inverted repeat (IR) boundary comparison for the seven *Asparagus* cpDNA genomes.**
Numbers in bold indicate the size of the gene (or gene section) within the specified regions. The numbers next to the dashed arrows indicate distances from the specified junctions. Numbers within the coloured bands indicate the lengths of the respective regions. The direction of gene transcription is presented by the obtuse angles of the pentagons. Ψ, pseudogene. Not to scale.

of 210 bp for *A. aethiopicus*, *A. densiflorus* 'Myers', and *A. racemosus*, and 279 bp for the other four species; it extended by 263–332 bp from the LSC/IR$_B$ junction into the IR$_B$.

The *ycf1* pseudogenes was retained in the border IR$_B$/SSC for all species, except *A. cochinchinensis* and *A. officinalis*; its length was 912 bp for all species except *A. schoberioides*, in which a 110 bp fragment of the SSC was deleted. *ndhF* in the SSC was 2229 bp long for *A. aethiopicus* and *A. densiflorus* 'Myers', and 2223 bp long for the other species; it extended from IR$_B$/SSC junction by 3 bp for *A. aethiopicus* and *A. densiflorus* 'Myers', 7 bp for *A. cochinchinensis*, and 9 bp for the others.

Functional *ycf1* genes (5624–5460 bp long) were located at the SSC/IR$_A$ border for all species except *A. officinalis*, in which an IR$_A$ portion was lost to the SSC, leaving a contracted pseudogene of 3824 bp in length. Further, in *A. setaceus*, the functional *ycf1* extended into the SSC by 307 bp from the SSC/IR$_A$ junction, unlike in the other species.

At the IR$_A$/LSC border, *rps19* (137–279 bp long) in IR$_A$ extended by 32–196 bp from the junction, with *A. offcinalis* having the shortest extension as a contracted pseudogene.

In the sliding-window analysis, five regions—*trnS-trnG*, *ndhC-trnV*, *accD-psaI*, *ccsA*, and *ycf1*—were identified as divergence hotspots with Pi ≥ 0.015 (Fig 9). *accD-psaI* was the most variable (Pi = 0.023), followed by *ccsA* (Pi = 0.020), and *trnS-trnG* (Pi = 0.17). These regions represent potential molecular markers for the phylogenetic and population genetics studies of *Asparagus* species. The sequence identity plot, using *A. aethiopicus* as a reference (S2 Fig), revealed different identity level (of <50%) among these five regions between the seven species, with "cracks" among the bars.

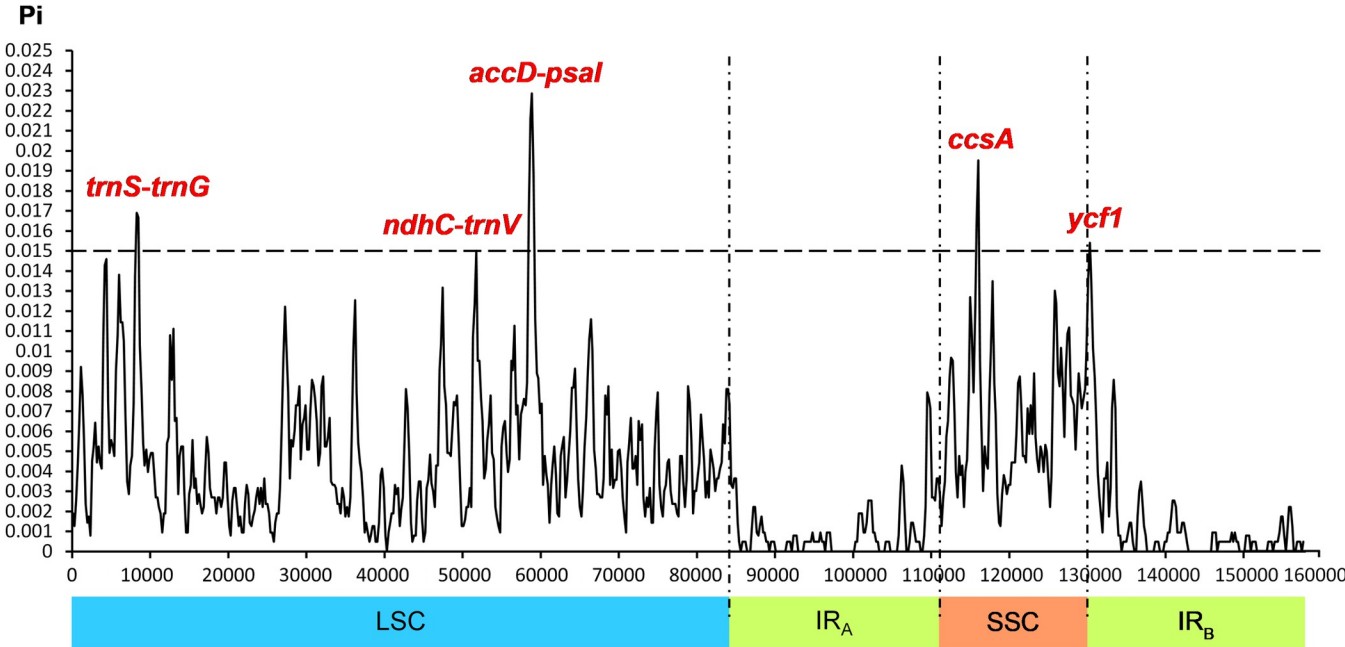

**Fig 9. Complete cpDNA genome nucleotide diversity for the seven *Asparagus* species.** X-axis: window midpoint; Y-axis: nucleotide diversity value (Pi) for each window. Divergence hotspots (Pi > 0.015) are labelled in red above the corresponding position.

Gene order and gene content were highly conserved among the seven species. The sequence identity plot (S2 Fig) revealed highly similar exon (purple) and intron (blue) regions. UTRs (red) in the non-coding regions clearly illustrate the diversity. The average Pi of 0.004 indicates that the sequence diversity of these species is relatively low.

No structural rearrangement was observed. IRs were more conserved than LSCs or SSCs, as illustrated by the high IR similarity in the sequence identity plot and supported by the sliding window analysis. The LSC and SSC regions contained most of the Pi peaks. In contrast, IRs had low nucleotide diversity (Pi < 0.01), except for the *ycf1* divergence hotspot at the SSC/IR border. The other four divergence hotspots were within LSCs (*trnS-trnG*, *ndhC-trnV*, and *accD-psaI*) and SSC (*ccsA*).

## Phylogenetic analysis

Congeneric relationships in the genus *Asparagus* were examined using three newly assembled cpDNA genomes and four cpDNA genomes from GenBank. ML trees derived from the complete cpDNA genomes, LSC, SSC, and CDS sequences shared the same topology (Fig 10) but different node bootstrap values. *A. setaceus* was sister to the other six *Asparagus* species. The branch containing *A. aethiopicus* and *A. densiflorus* 'Myers' had the highest bootstrap value (100) in all four ML trees, supporting the close relationship between these two species. *A. cochinchinensis* and *A. racemosus* formed a sister clade to *A. officinalis* and *A. schoberioides* (bootstrap values of 100 for complete cpDNA genomes, LSC, and SSC, and 84 for CDS). The close relationship between *A. cochinchinensis* and *A. racemosus* was well supported (bootstrap values of 100 for complete cpDNA genomes and LSC, and 99 for SSC and CDS). This new grouping differs from both traditional taxonomical classifications and molecular phylogenies [5, 6, 11]. We expected *A. racemosus*, a monoecious species, to group with the three other

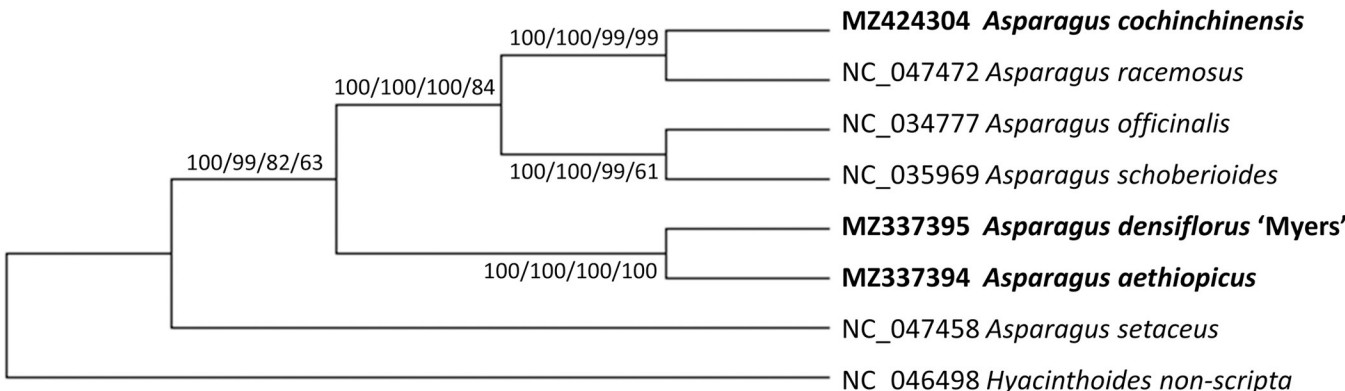

**Fig 10. Maximum likelihood (ML) trees based on *Asparagus* cpDNA genomes.** Numbers next to the nodes: bootstrap values based on complete cpDNA genomes/LSC/SSC/CDS sequences. The topologies are identical. Bold taxa: the three newly assembled cpDNA genomes.

monoecious species from South Africa. Instead, it was nested within the group of dioecious and Eurasian species in the ML trees, with high bootstrap values.

The ML tree based on IR sequences also exhibited unexpected grouping (Fig 11): *A. racemosus* was still nested with the dioecious species, which were sister to *A. officinalis* and *A. schoberioides*, with moderate support (bootstrap value = 71).

The close relationship between *A. aethiopicus* and *A. densiflorus* 'Myers' was supported by the ML trees based on complete cpDNA genomes, LSC, SSC, and CDS sequences (Fig 10) and was further validated by the IR-based tree, with bootstrap values of 100.

## Discussion

### Molecular insights for nomenclatural confusion

*A. aethiopicus* and *A. densiflorus* 'Myers' are nomenclaturally controversial. Batchelor and Scott (2006) [67] questioned the taxonomic identity of the cultivar 'Myers' (foxtail asparagus), which is often recorded as a cultivar of *A. densiflorus* [1, 3, 5, 11, 15, 51, 57, 64, 67, 80, 83]. In contrast, some have suggested placing the *Asparagus* cultivars 'Sprengeri' and 'Myers' under *A. aethiopicus* [4, 67, 76, 77]. The Royal Botanic Gardens Victoria [78, 108] has adopted the name *A. aethiopicus* 'Myersii' for foxtail asparagus.

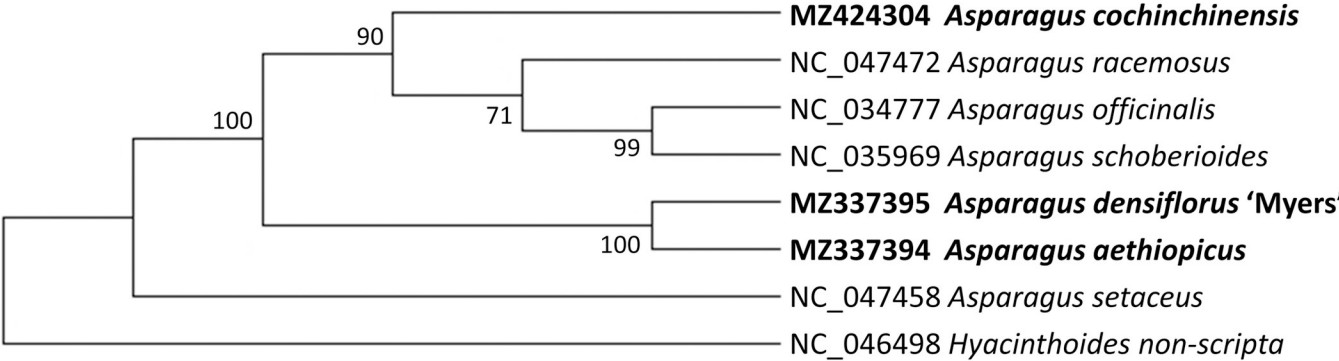

**Fig 11. Maximum likelihood (ML) trees based on inverted repeats (IRs) for *Asparagus*.** Numbers next to the nodes: bootstrap values on IR$_A$ and IR$_B$. Bold taxa: the three newly assembled cpDNA genomes.

*A. densiflorus* and *A. aethiopicus* differ primarily in their growth habit, with the former not a climber and rarely over 1 m tall and the latter an erect herb of 1 m or more and climbing up to 7 m [4, 76]. From our observations, the *Asparagus* cultivar 'Myers' never climbs, even when it is not pot-bound. This growth habit does not correspond with the circumscription of *A. aethiopicus* emphasised by Green (1986) [76] and Judd (2001) [4]. We agree with Batchelor and Scott (2006) that foxtail asparagus should be considered a cultivar of *A. densiflorus* [67], and hence the legitimate name should be *Asparagus densiflorus* (Kunth) Jessop 'Myers'.

Our findings show that *A. aethiopicus* and *A. densiflorus* 'Myers' are phylogenetically close, despite their morphological and growth habit differences, with bootstrap values of up to 100 for ML trees based on complete cpDNA genomes, LSC, SSC, IR, or CDS (Figs 10 and 11). Their gene numbers, GC content (Table 2), genome structure (Fig 2), and IR border (Fig 8) are similar. This supports the traditional classifications, which consistently place them under the same generic circumscription: genus *Asparagopsis* [71], genus *Asparagus* section *Falcati* [8], genus *Asparagus* section *Racemosi* [15], or genus *Protasparagus* [16] (S1 Fig and S1 Table). Using short-length DNA regions, Norup *et al.* [6] suggested placing the two species in an *Asparagus*–Racemose clade–Racemose 1 clade. Our phylogenetic results, which group the cpDNA genomes of *A. aethiopicus* and *A. densiflorus* 'Myers', are consistent with this.

The two species showed minor differences. In terms of LSR number and type, *A. densiflorus* 'Myers' differed significantly from *A. aethiopicus* and the other species. The cpDNA genome of *A. densiflorus* 'Myers' had the most LSRs, and this was the only species with reverse repeats and 40–49 bp LSRs (Figs 6 and 7). SSRs have been used to identify cultivars of potatoes [109, 110], apples [111], and sunflowers [112]. However, these *Asparagus* species did not differ significantly in SSRs. Nonetheless, the distinctive LSR patterns of *A. densiflorus* 'Myers' could provide a molecular authentication marker.

Our phylogenetic analysis revealed the close relationship between *A. aethiopicus* and *A. densiflorus* 'Myers' but did not elucidate the species origin of the cultivar. According to Article 21.1 of ICNCP, "*The name of a cultivar is a combination of the correct name of the genus or lower taxon to which it is assigned under the ICN, or its unambiguous common name, with a cultivar epithet*" [86]. We suggest two treatments to clarify *A. densiflorus* 'Myers' nomenclature: first, to combine only the genus name with the cultivar epithet, as *Asparagus* 'Myers', since this cultivar epithet has not been used for other cultivars being assigned to other *Asparagus* species; second, to combine the common name and the cultivar epithet, as asparagus 'Myers', since the common name of the genus *Asparagus* is unambiguous and is identical to the genus name.

## Unexpected placement of *A. racemosus*

Taxonomists have attempted to divide the genus *Asparagus* into three major groups. The first, characterised by flattened and leaf-like cladodes, basally connate perianth segments, and filaments connated into tubes, was classified as the genus *Myrsiphyllum* by Willdenow (1808) [14], Kunth (1850) [71], and Obermeyer (1984) [17], and as the genus *Asparagus* subgenus *Myrsiphyllum* by Baker (1875) [7]. The second and third groups comprise the species with filiform to linear cladodes: the second, comprising monoecious and African species with free perianth segments and filaments, was classified as the genus *Asparagopsis* by Kunth (1850) [71], the genus *Asparagus* subgenus *Asparagopsis* by Baker (1875) [7], and the genus *Protasparagus* by Obermeyer (1983) [17]; the third, comprising dioecious and Eurasian species with basally connate perianth segments, was classified as the genus *Asparagus* by Kunth (1850) [71] and the genus *Asparagus* subgenus *Euasparagus* by Baker (1875) [7].

*A. racemosus*, a monoecious species widespread throughout Africa, Asia, and Australia [10, 113], has traditionally been classified into the second group. Fukuda *et al.* [5] and Kubota *et al.*

[11] placed *A. racemosus* in genus *Asparagus* subgenus *Protasparagus*, whereas Norup *et al.* [6] placed it in the *Asparagus*–Racemose clade–Racemose 2 clade.

We expected *A. racemosus* to cluster with its relatives in the same group, i.e. *A. aethiopicus*, *A. densiflorus* 'Myers', and *A. setaceus*. However, one *A. racemosus* specimens (NC_047472) unexpectedly clustered with the dioecious species *A. cochinchinensis*, *A. officinalis*, and *A. schoberioides* in the ML trees (Figs 9 and 10), using both complete cpDNA genomes and sequence portions. This is contrary to Lee *et al.* (1997) [114] who, using restriction fragment length polymorphism cpDNA analysis, showed that no monoecious species were clustered within the monophyletic group of dioecious species (*A. officinalis*, *A. schoberiodes*, or *A. cochinchinensis*) [114].

Short cpDNA regions of *A. racemosus* (ca. 300–1000 bp) were reported by Fukuda *et al.* (*petB* intron and *petD-rpoA*) [5], Kubota *et al.* (*rpl32-trnL*, *trnQ-5′rps16*, *ndhF-rpl32*, *psbD-trnT*, *3′rps16-5′trnK*) [11], and Norup *et al.* (*3′ ndhF*, *psbA-trnH*, *trnD-trnT*) [6]. We attempted to determine the start and stop positions of these regions in NC_047472. Ten extracted sequences of the corresponding length (S2 Table) were screened using the NCBI Basic Local Alignment Search Tool, and only *trnQ-rps16* (sequence identity 98.70%), *psbA-trnH* (97.11% and 96.84%), *rpl32-trnL* (96.49%), *petD-rpoA* (96.86%), and *trnD-trnT* (97.71%) matched the respective regions of *A. racemosus*. GenBank did not contain any voucher information for NC_047472. Because of this lack of voucher information, we are unable to further verify this unanticipated and unlikely grouping. Our intra-generic analyses were constrained by the limited sample size. Further studies on *A. racemosus* phylogeny are recommended.

## Conclusion

Complete cpDNA genomes of three *Asparagus* specimens collected in Hong Kong were *de novo* assembled, annotated, and compared with those of congenerics. The seven genomes were relatively conserved in terms of gene content, gene order, and genome structure. *A. densiflorus* 'Myers' differed significantly from the others in LSR number and type. Five divergence hotspots were identified in the sliding-window analysis (Pi $\geq$ 0.015). Our phylogenetic analysis elucidates the generic subdivision and the nomenclatural complexity of *A. aethiopicus* and *A. densiflorus* 'Myers'. The novel placement of *A. racemosus*, contrary to previous morphological and molecular classifications, requires further verification. We suggest two ICNCP-compliant names for *A. densiflorus* 'Myers', namely *Asparagus* 'Myers' and asparagus 'Myers'. These *de novo* assembled cpDNA genomes provide potential genomic resources, elucidating *Asparagus* taxonomy, application, and conservation.

## Supporting information

**S1 Fig. The historical changes on the generic subdivision of the genus *Asparagus*.**
(PDF)

**S2 Fig. Visualisation of the alignments of 7 *Asparagus* chloroplast genomes using *A. aethiopicus* as a reference.**
(PDF)

**S3 Fig. Specimen photos of voucher K. H. Wong 092, 107, and 109.**
(PDF)

**S1 Table. The historical changes on taxonomical status of the 7 studied *Asparagus* species.**
(XLSX)

**S2 Table. Extracted sequences from cpDNA of *Asparagus racemosus* (NC_047472.1).**
(XLSX)

## Author Contributions

**Conceptualization:** Kwan-Ho Wong.

**Data curation:** Kwan-Ho Wong, Bobby Lim-Ho Kong.

**Formal analysis:** Kwan-Ho Wong.

**Funding acquisition:** Pang-Chui Shaw, David Tai-Wai Lau.

**Investigation:** Kwan-Ho Wong.

**Methodology:** Bobby Lim-Ho Kong.

**Project administration:** Pang-Chui Shaw, David Tai-Wai Lau.

**Resources:** Pang-Chui Shaw, David Tai-Wai Lau.

**Software:** Bobby Lim-Ho Kong.

**Supervision:** Pang-Chui Shaw, David Tai-Wai Lau.

**Validation:** Bobby Lim-Ho Kong, Tin-Yan Siu, Hoi-Yan Wu, David Tai-Wai Lau.

**Visualization:** Kwan-Ho Wong.

**Writing – original draft:** Kwan-Ho Wong.

**Writing – review & editing:** Tin-Yan Siu, Hoi-Yan Wu, Grace Wing-Chiu But.

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
