## [Decision Letter · Decision Letter 0]

17 Nov 2021

PONE-D-21-33675Complete chloroplast genomes of *Asparagus aethiopicus* L., *Asparagus densiflorus* (Kunth) Jessop ‘Myers’ & *Asparagus cochinchinensis* (Lour.) Merr.: Comparative and phylogenetic analysis with congeneric speciesPLOS ONE

Dear Dr. LAU,

Thank you for submitting your manuscript to PLOS ONE. After careful consideration, we feel that it has merit but does not fully meet PLOS ONE’s publication criteria as it currently stands. Therefore, we invite you to submit a revised version of the manuscript that addresses the points raised during the review process.

As suggested by Reviewer #1, literature update is highly recommended, as the authors did not present the latest findings related to the scientific matter. Moreover, an in depth authors' elaboration of the present text regarding the literature update is expected.

The whole text would hugely benefit if proofread by a native English speaker or a professional editing agency. Some but not all places of much needed intervention are following:

L36: "species" should not stand italicized.

L39: "Asparagus have been widely applied..." - a colloquial expression. Suggestion: "Many species belonging to the Asparagus genus have been widely applied..."

L41, L58 and elsewhere in the text: "anthropocentric usage of Asparagus", "while some Asparagus were used"... - the same comment as the previous.

L47 and further in the text: Please use "A." to abbreviate the genus name when introducing a species name.

L77 and elsewhere in the text: Do not capitalize common plant names such as "Garden Asparagus". Write "garden asparagus".

Please note that Reviewer #2 has stated "...I have labelled these shortcomings in this paper;..." but did not provide the attachment where these shortcomings might be visible. The reviewer remains unresponsive regarding this issue.

We look forward to receiving your revised manuscript.

Kind regards,

Branislav T. Šiler, Ph.D.

Academic Editor

PLOS ONE

Journal Requirements:

Reviewers' comments:

Reviewer's Responses to Questions

**Comments to the Author**

1. Is the manuscript technically sound, and do the data support the conclusions?

Reviewer #1: Yes

Reviewer #2: Yes

2. Has the statistical analysis been performed appropriately and rigorously? 

Reviewer #1: Yes

Reviewer #2: Yes

3. Have the authors made all data underlying the findings in their manuscript fully available?

Reviewer #1: Yes

Reviewer #2: Yes

4. Is the manuscript presented in an intelligible fashion and written in standard English?

Reviewer #1: Yes

Reviewer #2: Yes

5. Review Comments to the Author

Reviewer #1: The Authors have carried out a study based on the sequencing of the Complete Chloroplast Genome of three species (A. aethiopicus, A. densiflorus ‘Myers’ and A. cochinchinensis) belonging to the Asparagus genus. The results obtained were assembled with chloroplast genomes of other four Asparagus species (A. setaceus, A. racemosus, A. schoberioides Asparagus officinalis L.) on NCBI. The manuscript could bring a significative impact for further studies aimed to widen the knowledge of the Asparagus genus and the phylogenetic relationships among Asparagus spp. In my opinion, the paper could be published after a few modifications that might improve this manuscript.

The main criticism to this paper is that I have lacked the mention of some studies employing cpDNA that have been previously developed in the Asparagus genus ( Lee et al 1996; Lee et al 1997; Kanno et al 1997; Seng et al 2017 and Li et al 2019). I was impressed about the comprehensive and detailed revision on the taxonomic classifications in the asparagus genus that have been published since the eighteen century to date. However, previous studies employing cpDNA of different Asparagus spp. are not mentioned in the manuscript. In my opinion these studies should be mentioned in the Introduction section of the manuscript. I think that at least the studies developed by Lee et al. (1997) and Seng et al. (2017) are recommended to be also used in the discussion of the manuscript because the findings of the present study can be compared to a certain extent with the obtained by these two studies.

Other comments and suggestions for the Authors:

> Abstract: The authors wrote: “Conducting comparative and phylogenetic analysis with congeneric species, four cpDNA on NCBI were included in this study”. The scientific name of these four species (A. setaceus, A. racemosus, A. schoberioides Asparagus officinalis L.) they should be included in this section of the manuscript.

> lines 35-37: In these lines is written the following sentence: “These aforementioned characteristics were evolved by Asparagus species to adapt to arid environment”. I think that this sentence must be support by a cite/s

> lines 43-44: It is written: “of the two studied Asparagus species were discussed in detail” Is it correct? Two (A. aethiopicus, A. densiflorus ‘Myers’) or three species (A. aethiopicus, A. densiflorus ‘Myers’, A. cochinchinensis)?

> Lines 78- 79. The authors wrote: “However, the gene pool of A. officinalis is relatively limited” In my opinion there are other studies such as Geoffriau et al 1992, Moreno et al 2006 or Mercati et al 2015 that must be cited instead of the study carried out by Stajner et al 2002

> lines 79-80: I suggest modifying the following sentence: “The species is susceptible to multiple diseases” by the species is susceptible to multiple biotic and abiotic stresses…

Lee, Y. O., Kanno, A., & Kameya, T. (1997). Phylogenetic relationships in the genus Asparagus based on the restriction enzyme analysis of the chloroplast DNA. Japanese Journal of Breeding, 47(4), 375-378.

Sheng, W., Chai, X., Rao, Y., Tu, X., & Du, S. (2017). Complete chloroplast genome sequence of Asparagus (asparagus officinalis l.) and its phylogenetic position within asparagales. Journal of Plant Breeding and Genetics, 5(3), 121-128.

Lee, Y. O., Kanno, A., & Kameya, T. (1996). The physical map of the chloroplast DNA from Asparagus officinalis L. Theoretical and Applied Genetics, 92(1), 10-14.

Kanno, A., Lee, Y. O., & Kameya, T. (1997). The structure of the chloroplast genome in members of the genus Asparagus. Theoretical and applied genetics, 95(8), 1196-1202.

Li, J. R., Li, S. F., Wang, J., Dong, R., Zhu, H. W., Li, N., ... & Gao, W. J. (2019). Characterization of the complete chloroplast genome of Asparagus setaceus. Mitochondrial DNA Part B, 4(2), 2639-2640.

Moreno, R., Espejo, J. A., Cabrera, A., Millan, T., & Gil, J. (2006). Ploidic and molecular analysis of ‘Morado de Huetor’asparagus (Asparagus officinale L.) population; a Spanish tetraploid landrace. Genetic Resources and Crop Evolution, 53(4), 729-736.

Geoffriau E, Denoue D, Rameau C (1992) Assessment of genetic variation among asparagus (Asparagus officinalis L.) populations and cultivars: agromorphological and isozymic data. Euphytica 61(3):169–179

Mercati F, Riccardi P, Harkess A et al (2015) Single nucleotide polymorphism–based parentage analysis and population structure in garden asparagus, a worldwide genetic stock classification. Mol Breed 35(2):59.

Reviewer #2: The introduction part is chaotic in this manuscript, which shoule be rewritten；Please pay attention to some mistakes in your grammar and spelling, and I have labelled these shortcomings in this paper; the content is complete, and the result is clear; and I suggest this paper should be accepted after minor revison.

6. PLOS authors have the option to publish the peer review history of their article (what does this mean?). If published, this will include your full peer review and any attached files.

Reviewer #1: No

Reviewer #2: **Yes: **Wentao Sheng

---

## [Author Response · Author response to Decision Letter 0]

28 Jan 2022

Dear Dr. Šiler, Reviewer 1 and Dr. Sheng,

Rebuttal Letter

I am sincerely writing to you to response all your comments in the Decision Letter in 18th November 2021. Our replies are listed as below.

Comment 1. As suggested by Reviewer #1, literature update is highly recommended, as the authors did not present the latest findings related to the scientific matter. Moreover, an in depth authors' elaboration of the present text regarding the literature update is expected.

Reply: Agreed. We have added the relevant articles in the reference list and elaborated their findings in the sections of introduction, result and discussion.

Comment 2. The whole text would hugely benefit if proofread by a native English speaker or a professional editing agency…

Reply: Agreed. We will employ a professional editing agency to proofread the article after all parties agreed with the content.

Comment 3. L36: "species" should not stand italicized.

Reply: Agreed and revised.

Comment 4. L39: "Asparagus have been widely applied..." - a colloquial expression. Suggestion: "Many species belonging to the Asparagus genus have been widely applied..."

Reply: Agreed. Amendment was made as follows:

“Many species belonging to the genus Asparagus have been widely applied in different aspects of human society”. 

The genus name was italicized and put after the word “genus” according to the rule of ICN.

Comment 5. L41, L58 and elsewhere in the text: "anthropocentric usage of Asparagus", "while some Asparagus were used"... - the same comment as the previous.

Reply: Agreed and amended as follows:

L60-61: Adapting to drought conditions, Asparagus species have evolved characteristic morphology.

L71-72: Below we briefly discussed the anthropocentric usage of some Asparagus species…

L89: …while some Asparagus species were used as charm to increase fertility…

Comment 6. L47 and further in the text: Please use "A." to abbreviate the genus name when introducing a species name.

Reply: We agreed with the guidelines of PLOS One. We had adopted abbreviated genus name (A.) for most of the binominal names in the article. However, we would like to keep the presentation of full scientific name for the sentences below, in order to (i) obey the rules of ICN/ICNCP, (ii) accurately present the nomenclatural treatment, and (iii) present the original record of literatures.

L165-167: In 1890, Regel published the name Asparagus sprengeri based on the cultivated plants growing in Natal

L168-169: The name Asparagus sprengeri Regel hence is the scientific name of sprengeri asparagus. [Remarks: the species epithets “sprengeri” is amended in lower case instead of “Sprengeri” in the first submission]

L171-172: Jessop synonymized A. sprengeri Regel under the new combination Asparagus densiflorus (Kunth) Jessop based on morphology and geographical distribution

L178-179: In 1767, Linnaeus published the name Asparagus aethiopicus L. in the Species Plantarum

L187-188: Both names Aspararagopsis aethiopica (and later, Asparagus aethiopicus) and Asparagopsis densiflora were adopted in parallel for 116 years, from 1850 to 1965.

L197-200: Straley and Utech also adopted the scientific name A. aethiopicus for sprengeri asparagus, and stated “Asparagus densiflorus (Kunth) Jessop has been misapplied to this species” in Flora of North America North of Mexico (2004)

L207-210: …in terms of its habitats, growing habits and reproductive characters, fits the circumscription of Asparagus aethiopicus L. in the monograph. Therefore, we adopt the scientific name Asparagus aethiopicus L. for sprengeri asparagus throughout this study.

L218-220: The first binomial name of foxtail asparagus, Asparagus myersii, was raised anonymously in an unknown time, while the name Asparagopsis densiflora was validly published in 1850 by Kunth (S2 Table)

L222-223: In 1966, Jessop mentioned the name Asparagus myersii Hort. “had never been validly published”

L239-241: …, we follow the treatment of some taxonomists and scientists [1,4-5,11,51,64,76,83], adopting the scientific name Asparagus densiflorus Jessop (Kunth) ‘Myers’ for foxtail asparagus throughout this study. 

L306: Remains the full name of Asparagus aethiopicus L., Asparagus densiflorus (Kunth) Jessop 'Myers' and Asparagus cochinchinensis (Lour.) Merr. were used in Table 1 which indicates the authentication of specimens.

L614-616: We agreed with Batchelor & Scott (2006) that foxtail asparagus should be a cultivar of A. densiflorus [67], and hence the legitimate name should be Asparagus densiflorus (Kunth) Jessop ‘Myers’.

L647-648: One is to combine only the genus name with the cultivar epithet, as Asparagus ‘Myers’…

L649-651: Another treatment is the combination of the common name and the cultivar epithet, as Asparagus ‘Myers’… [In this case “Asparagus” is not italic and not a genus name]

Comment 7. L77 and elsewhere in the text: Do not capitalize common plant names such as "Garden Asparagus". Write "garden asparagus".

Reply: Agreed. All are amended. 

Comments from Reviewer 1:

The Authors have carried out a study based on the sequencing of the Complete Chloroplast Genome of three species (A. aethiopicus, A. densiflorus ‘Myers’ and A. cochinchinensis) belonging to the Asparagus genus. The results obtained were assembled with chloroplast genomes of other four Asparagus species (A. setaceus, A. racemosus, A. schoberioides Asparagus officinalis L.) on NCBI. The manuscript could bring a significative impact for further studies aimed to widen the knowledge of the Asparagus genus and the phylogenetic relationships among Asparagus spp. In my opinion, the paper could be published after a few modifications that might improve this manuscript.

The main criticism to this paper is that I have lacked the mention of some studies employing cpDNA that have been previously developed in the Asparagus genus ( Lee et al 1996; Lee et al 1997; Kanno et al 1997; Seng et al 2017 and Li et al 2019). I was impressed about the comprehensive and detailed revision on the taxonomic classifications in the asparagus genus that have been published since the eighteen century to date. However, previous studies employing cpDNA of different Asparagus spp. are not mentioned in the manuscript. In my opinion these studies should be mentioned in the Introduction section of the manuscript. I think that at least the studies developed by Lee et al. (1997) and Seng et al. (2017) are recommended to be also used in the discussion of the manuscript because the findings of the present study can be compared to a certain extent with the obtained by these two studies.

Reply: Thank you so much for your appreciation! We agreed with your suggestions and advices. The suggested literatures were already included in the sections of introduction, result and discussion with appropriate elaboration. 

 

Other comments and suggestions for the Authors:

> Abstract: The authors wrote: “Conducting comparative and phylogenetic analysis with congeneric species, four cpDNA on NCBI were included in this study”. The scientific name of these four species (A. setaceus, A. racemosus, A. schoberioides Asparagus officinalis L.) they should be included in this section of the manuscript.

Reply: Agreed. The scientific names of the four species were included accordingly.

> lines 35-37: In these lines is written the following sentence: “These aforementioned characteristics were evolved by Asparagus species to adapt to arid environment”. I think that this sentence must be support by a cite/s

Reply: Agreed. The reference source (Dahlgren et al., 1985 and Judd, 2001) were cited accordingly.

> lines 43-44: It is written: “of the two studied Asparagus species were discussed in detail” Is it correct? Two (A. aethiopicus, A. densiflorus ‘Myers’) or three species (A. aethiopicus, A. densiflorus ‘Myers’, A. cochinchinensis)?

Reply: Agreed. The word “two” was deleted from this sentence.

> Lines 78- 79. The authors wrote: “However, the gene pool of A. officinalis is relatively limited” In my opinion there are other studies such as Geoffriau et al 1992, Moreno et al 2006 or Mercati et al 2015 that must be cited instead of the study carried out by Stajner et al 2002

Reply: Agreed. We cited your suggested source of literature instead of the one of Stajner et al., 2002.

> lines 79-80: I suggest modifying the following sentence: “The species is susceptible to multiple diseases” by the species is susceptible to multiple biotic and abiotic stresses…

Reply: Agreed. Amendment was done.

Comments from Reviewer 2:

The introduction part is chaotic in this manuscript, which shoule be rewritten；Please pay attention to some mistakes in your grammar and spelling, and I have labelled these shortcomings in this paper; the content is complete, and the result is clear; and I suggest this paper should be accepted after minor revison.

Reply: Thank you for your comments. We have rewritten the introduction. For the grammar and spelling, we have revised it accordingly, and will sent the manuscript to a professional agency for further proofread after all parties agreed with the content. 

Other comments and suggestions for the Authors:

Line 72. Hong Kongwhere

Reply: Agreed and amended. Space between “Kong” and “where” was added.

Line 233. S2 Table

Reply: The sentence in Line 233 had been deleted in the latest version. The usage of S2 Table in other parts of the manuscript were put in brackets. 

Line 354. The pseudogene ycf1...

Reply: Agreed. The gene name (ycf1) was italicized.

Line 471. ranging from 5624 to 5460 bp ...

Reply: Agreed. The unit was given for each number as “5624 bp to 5460 bp”.

Line 481. trnS-trnG (Pi = 0.17)

Reply: Agreed. The sentence was revised as “Among these hotspots, accD-psaI was the most variable region (Pi=0.023), followed by ccsA (Pi=0.020) and then trnS-trnG (Pi = 0.17).”

Line 549. [4,Error! Reference source not found.,65,74].

Reply: Agreed. The reference source (Straley & Utech, 2003) was updated.

Line 604. The second group consisting…

Reply: Agreed. The sentence was revised as "The second group consisting of".

Line 628. S4 Table

Reply: Sorry. We have no idea of this amendment. Please further advice. 

 

Comments from the Academic Editor:

Reply: Agreed. We have updated our manuscript according to the style requirements.

Reply: The research work was supported by a donation fund from Wu Jieh Yee Charitable Foundation Limited. The fund has no formal grant number. 

Reply: Agreed. This phrase was removed as the data are not a core part of the research.

 

Reply: 

The following references were newly added along with the amended manuscript:

38. Geoffriau E, Denoue D, Rameau C. Assessment of genetic variation among asparagus (Asparagus officinalis L.) populations and cultivars: agromorphological and isozymic data. Euphytica. 1992; 61:169-179. doi:10.1007/BF00039655

39. Mercati F, Riccardi P, Harkess A. Single nucleotide polymorphism-based parentage analysis and population structure in garden asparagus, a worldwide genetic stock classification. Mol Breeding. 2015. 35(59):1-12. doi:10.1007/s11032-015-0217-5

40. Moreno R, Espejo JA, Cabrera A, Millán T, Gil J. Ploidic and Molecular Analysis of ‘Morado de Huetor’ asparagus (Asparagus officinalis L.) population; a Spanish tetraploid landrace. 2006; 53:729–736. Genet Resour Crop Evol. doi:10.1007/s10722-004-4717-0

82. Obermeyer AA, Immelamn KL, Bos JJ. (1992). Asparagaceae. In: Leistner OA, du Plessis E, editors. Flora of Southern Africa. Volumn 5, Part 3. Dracaenaceae, Asparagaceae, Luzuriagaceae and Smilacaceae. Pretoria: National Botanical Institute; 1992. pp. 11-82.

93. Lee YO, Kanno A, Kameya T. The physical map of the chloroplast DNA from Asparagus officinalis L. Theor Appl Genet. 1996;92:10-14.

94. Kanno A, Lee YO, Kameya T. The structure of the chloroplast genome in members of the genus Asparagus. Theor Appl Genet. 1997;95:1196-1202.

95. Sheng W, Chai X, Rao Y, Tu X & Du S. Complete chloroplast genome sequence of Asparagus (Asparagus officinalis L.) and its phylogenetic position within Asparagales. J Plant Breed Genet. 2017; 5(3):121-128.

96. Li JR, Li SF, Wang J, Dong R, Zhu HW, Li N, et al. Characterization of the complete chloroplast genome of Asparagus setaceus. Mitochondrial DNA B Resour, 2019; 4(2):2639-2640. doi: 10.1080/23802359.2019.1643798

114. Lee YO, Kanno A, Kameya T. Phylogenetic relationships in the genus Asparagus based on the restriction enzyme analysis of the chloroplast DNA. Japanese Journal of Breeding. 1997; 47:375-378.

The following reference was deleted according to the comment from Reviewer 1:

45. Stajner N, Bohanec B, Javornik B. Genetic variability of economically important Asparagus species as revealed by genome size analysis and rDNA ITS polymorphisms. Plant Sci. 2002; 162(6):931-937. doi: 10.1016/S0168-9452(02)00039-0

Thank you very much for your kindly reviews and consideration. 

Yours sincerely,

Dr. David TW LAU

Curator of the Shiu-ying Hu Herbarium

School of Life Sciences

The Chinese University of Hong Kong

---

## [Editor Report · Decision Letter 1]

31 Jan 2022

PONE-D-21-33675R1Complete chloroplast genomes of *Asparagus aethiopicus* L., *Asparagus densiflorus* (Kunth) Jessop ‘Myers’ & *Asparagus cochinchinensis* (Lour.) Merr.: Comparative and phylogenetic analysis with congeneric speciesPLOS ONE

Dear Dr. LAU,

Thank you for submitting your manuscript to PLOS ONE. After careful consideration, we feel that it has merit but does not fully meet PLOS ONE’s publication criteria as it currently stands. Therefore, we invite you to submit a revised version of the manuscript that addresses the points raised during the review process.

Following the rules of ICN (not ICNCP, since the studied taxa are not cultivated) is absolutely necessary in scientific literature and I fully support it. However, please bear in mind that abbreviating genus name does not oppose the ICN rules (the current version is Shenzhen Code, published by IAPT - https://www.iapt-taxon.org/nomen/main.php; for genera, please see https://www.iapt-taxon.org/nomen/pages/main/art_20.html). It is a common scientific practice to write a scientific name in full when it is first used or when several species from the same genus are being listed or discussed in the same paper or report. For subsequent uses, the genus can be abbreviated to its first letter followed by a period. You can abbreviate the genus name after its first use even when describing a different species within that genus, as long as there is no risk of confusing it for another genus or genera (which is not the case here, since only the genus *Asparagus* circulates throughout the manuscript). Therefore, by abbreviating the genus name (i) obeying the rules of ICN is not compromised, ii) nomenclatural treatment is presented accurately (no confusion to other genera starting with "*A*."), and iii) if a species is described in a source article as e.g., "*Asparagus aethiopicus*", I do not see the reason why in consequent articles dealing with the same species it wouldn't be abbreviated as "*A. aethiopicus*" if the genus name was already mentioned earlier in the text. The same applies to the main title: reprising the genus name three times is space-consuming and quite exhausting for reading. I suggest abbreviating here the genus name for the second and the third species as well (also, replace "&" with "and" here and elsewhere in the text). The exceptions might be L178-179 (cites the original text), L197-200 (cites the original text), L218-220 (only for mentioning *Asparagus myersii*, since a binomial name has been announced), L222-223 (cites the original text), Table 1, L614-616 (a full name has been announced), L647-648 (as it complies with the ICN rules), L649-651 (but being the common name, "asparagus” should stand in lowercase).

Moreover, vernacular expressions such as L101 (circumscription of *Asparagus*), L117 (relationships within *Asparagus*), L125 (only one native *Asparagus*), L126 (Other common exotic *Asparagus*), L232 (7 *Asparagus *were examined) (and in many other places) were not clarified as required in the previous review round.

102-107: The terms "genus" and "subgenus" should stand in lowercase.

L127-129 and elsewhere in the text: I must repeat: common names should be written in lowercase, i.e., "sprengeri asparagus" (however, I'm not aware of this Latinized common name, but "Sprenger's asparagus" might be acceptable, after Carl Ludwig Sprenger), "lace fern", etc.

The authors are urged to meticulously check the text once again for proper English usage while following the comments stated above. I still strongly encourage the authors to have the MS proofread by a native English speaker or professional editing agency.

We look forward to receiving your revised manuscript.

Kind regards,

Branislav T. Šiler, Ph.D.

Academic Editor

PLOS ONE
---

## [Author Response · Author response to Decision Letter 1]

16 Mar 2022

Dear Dr. Šiler,

Rebuttal Letter

I am sincerely writing to you to response all your comments in the Decision Letter in 31st January 2022. Our replies are listed as below.

Original comments from Dr. Šiler:

Following the rules of ICN (not ICNCP, since the studied taxa are not cultivated) is absolutely necessary in scientific literature and I fully support it. However, please bear in mind that abbreviating genus name does not oppose the ICN rules (the current version is Shenzhen Code, published by IAPT - https://www.iapt-taxon.org/nomen/main.php; for genera, please see https://www.iapt-taxon.org/nomen/pages/main/art_20.html). It is a common scientific practice to write a scientific name in full when it is first used or when several species from the same genus are being listed or discussed in the same paper or report. For subsequent uses, the genus can be abbreviated to its first letter followed by a period. You can abbreviate the genus name after its first use even when describing a different species within that genus, as long as there is no risk of confusing it for another genus or genera (which is not the case here, since only the genus Asparagus circulates throughout the manuscript). Therefore, by abbreviating the genus name (i) obeying the rules of ICN is not compromised, ii) nomenclatural treatment is presented accurately (no confusion to other genera starting with "A."), and iii) if a species is described in a source article as e.g., "Asparagus aethiopicus", I do not see the reason why in consequent articles dealing with the same species it wouldn't be abbreviated as "A. aethiopicus" if the genus name was already mentioned earlier in the text. The same applies to the main title: reprising the genus name three times is space-consuming and quite exhausting for reading. I suggest abbreviating here the genus name for the second and the third species as well (also, replace "&" with "and" here and elsewhere in the text). The exceptions might be L178-179 (cites the original text), L197-200 (cites the original text), L218-220 (only for mentioning Asparagus myersii, since a binomial name has been announced), L222-223 (cites the original text), Table 1, L614-616 (a full name has been announced), L647-648 (as it complies with the ICN rules), L649-651 (but being the common name, "asparagus” should stand in lowercase).

Moreover, vernacular expressions such as L101 (circumscription of Asparagus), L117 (relationships within Asparagus), L125 (only one native Asparagus), L126 (Other common exotic Asparagus), L232 (7 Asparagus were examined) (and in many other places) were not clarified as required in the previous review round.

102-107: The terms "genus" and "subgenus" should stand in lowercase.

L127-129 and elsewhere in the text: I must repeat: common names should be written in lowercase, i.e., "sprengeri asparagus" (however, I'm not aware of this Latinized common name, but "Sprenger's asparagus" might be acceptable, after Carl Ludwig Sprenger), "lace fern", etc.

The authors are urged to meticulously check the text once again for proper English usage while following the comments stated above. I still strongly encourage the authors to have the MS proofread by a native English speaker or professional editing agency.

Reply: Thank you so much for your detailed clarification. Please see the reply of the subtracted comments from your comments:

Comment 1: Therefore, by abbreviating the genus name (i) obeying the rules of ICN is not compromised, ii) nomenclatural treatment is presented accurately (no confusion to other genera starting with "A."), and iii) if a species is described in a source article as e.g., "Asparagus aethiopicus", I do not see the reason why in consequent articles dealing with the same species it wouldn't be abbreviated as "A. aethiopicus" if the genus name was already mentioned earlier in the text.

Reply: Agreed. Please see the amendment in the manuscript.

Comment 2: The same applies to the main title: reprising the genus name three times is space-consuming and quite exhausting for reading. I suggest abbreviating here the genus name for the second and the third species as well …

Reply: Agreed. Please see the amendment in the manuscript.

Comment 3: (also, replace "&" with "and" here and elsewhere in the text)

Reply: Agreed. All the sign “&” were placed by the word “and” in the main text and title.

Comment 4: The exceptions might be L178-179 (cites the original text), L197-200 (cites the original text), L218-220 (only for mentioning Asparagus myersii, since a binomial name has been announced), L222-223 (cites the original text), Table 1, L614-616 (a full name has been announced), L647-648 (as it complies with the ICN rules), L649-651 (but being the common name, "asparagus” should stand in lowercase).

Reply: Thank you so much for your consideration for these exception. However, two more exceptions we would like to request:

L165-167: In 1890, Regel published the name Asparagus sprengeri based on the cultivated plants growing in Natal. 

Reason: Cites the original text

L187-188: Both names Aspararagopsis aethiopica (and later, Asparagus aethiopicus) and Asparagopsis densiflora were adopted in parallel for 116 years, from 1850 to 1965. 

Reason: Please noticed that there are two genera involved: Asparagopsis and Asparagus. Both the abbreviation of these two genera are “A.” which would be confusing. Moreover, please noticed that Asparagopsis aethiopica and Asparagopsis densiflora were the name published by Kunth in 1850 (Please refer to the S2 Table). 

Comment 5: Moreover, vernacular expressions such as L101 (circumscription of Asparagus), L117 (relationships within Asparagus), L125 (only one native Asparagus), L126 (Other common exotic Asparagus), L232 (7 Asparagus were examined) (and in many other places) were not clarified as required in the previous review round.

Reply: Agreed. Please see the amendment in the manuscript.

Comment 6: 102-107: The terms "genus" and "subgenus" should stand in lowercase.

Reply: Agreed. Please see the amendment in the manuscript.

Comment 7: L127-129 and elsewhere in the text: I must repeat: common names should be written in lowercase, i.e., "sprengeri asparagus" (however, I'm not aware of this Latinized common name, but "Sprenger's asparagus" might be acceptable, after Carl Ludwig Sprenger), "lace fern", etc.

Reply: Agreed. The common name “Sprenger’s asparagus” is adopted.

Comment 8: The authors are urged to meticulously check the text once again for proper English usage while following the comments stated above. I still strongly encourage the authors to have the MS proofread by a native English speaker or professional editing agency.

Reply: Agreed. We employed a professional editing agency to proofread the manuscript.

Thank you very much for your kindly reviews and consideration. 

Yours sincerely,

Dr. David TW LAU

Curator of the Shiu-ying Hu Herbarium

School of Life Sciences

The Chinese University of Hong Kong

---

## [Editor Report · Decision Letter 2]

21 Mar 2022

Complete chloroplast genomes of *Asparagus aethiopicus* L., *A. densiflorus* (Kunth) Jessop ‘Myers’, and *A. cochinchinensis* (Lour.) Merr.: Comparative and phylogenetic analysis with congenerics

PONE-D-21-33675R2

Dear Dr. LAU,

We’re pleased to inform you that your manuscript has been judged scientifically suitable for publication and will be formally accepted for publication once it meets all outstanding technical requirements.

Kind regards,

Branislav T. Šiler, Ph.D.

Academic Editor

PLOS ONE
---

## [Editor Report · Acceptance letter]

31 Mar 2022

PONE-D-21-33675R2 

Complete chloroplast genomes of *Asparagus aethiopicus* L., A. *densiflorus* (Kunth) Jessop ‘Myers’, and A. *cochinchinensis* (Lour.) Merr.: Comparative and phylogenetic analysis with congenerics 

Dear Dr. LAU:

I'm pleased to inform you that your manuscript has been deemed suitable for publication in PLOS ONE. Congratulations! Your manuscript is now with our production department. 

Kind regards, 

on behalf of

Dr. Branislav T. Šiler 

Academic Editor

PLOS ONE